# Suggestion of creatine as a new neurotransmitter by approaches ranging from chemical analysis and biochemistry to electrophysiology

Xiling Bian[1,2†], Jiemin Zhu[1,2†], Xiaobo Jia[1,2†], Wenjun Liang[3,4], Sihan Yu[1,4], Zhiqiang Li[1], Wenxia Zhang[1,3,5], Yi Rao[1,2,3,4,5,6]*

[1]Laboratory of Neurochemical Biology, PKU-IDG/McGovern Institute for Brain Research, Peking-Tsinghua Center for Life Sciences, School of Life Sciences, Department of Chemical Biology, College of Chemistry and Chemical Engineering, Department of Molecular and Cellular Pharmacology, School of Pharmaceutical Sciences, Peking University, Beijing, China; [2]Chinese Institute for Brain Research (CIBR), Beijing, China; [3]Chinese Institutes of Medical Research, Capital Medical University, Beijing, China; [4]Changping Laboratory, Yard 28, Science Park Road, Changping District, Beijing, China; [5]Institute of Molecular Physiology, Shenzhen Bay Laboratory, Shenzhen, China; [6]Research Unit of Medical Neurobiology, Chinese Academy of Medical Sciences, Beijing, China

*For correspondence:
yrao@pku.edu.cn

†These authors contributed equally to this work

Competing interest: The authors declare that no competing interests exist.

**Abstract** The discovery of a new neurotransmitter, especially one in the central nervous system, is both important and difficult. We have been searching for new neurotransmitters for 12 y. We detected creatine (Cr) in synaptic vesicles (SVs) at a level lower than glutamate and gamma-aminobutyric acid but higher than acetylcholine and 5-hydroxytryptamine. SV Cr was reduced in mice lacking either arginine:glycine amidinotransferase (a Cr synthetase) or SLC6A8, a Cr transporter with mutations among the most common causes of intellectual disability in men. Calcium-dependent release of Cr was detected after stimulation in brain slices. Cr release was reduced in *Slc6a8* and *Agat* mutants. Cr inhibited neocortical pyramidal neurons. SLC6A8 was necessary for Cr uptake into synaptosomes. Cr was found by us to be taken up into SVs in an ATP-dependent manner. Our biochemical, chemical, genetic, and electrophysiological results are consistent with the possibility of Cr as a neurotransmitter, though not yet reaching the level of proof for the now classic transmitters. Our novel approach to discover neurotransmitters is to begin with analysis of contents in SVs before defining their function and physiology.

## eLife assessment

This study presents **valuable** observations on a potential role of creatine (Cr) as a novel neurotransmitter. The data provide **solid** evidence that Cr is present in synaptic vesicles. If, in the future, a receptor can be described, it will support the claim that Cr is synaptically released and binds to a post-synaptic receptor. This would be of wide interest to the field of neuroscience.

## Introduction

Neural signaling depends on chemical transmission between neurons and their target cells (*Du Bois-Reymond, 1877*; *Langley, 1901b*; *Elliot, 1904*; *Elliott, 1905*; *Dale, 1914*; *Loewi, 1921*). Neurotransmission depends on chemicals such as neurotransmitters, neuromodulators, and neuropeptides. Decades of work, sometimes with convoluted paths, were involved before a molecule was established as a classic neurotransmitter (*Rao, 2019*). Initial hints about cholinergic signaling were obtained in the 1800s (*Bernard, 1857*; *Langley and Dickinson, 1890*; *Langley, 1905*; *Langley, 1906*). Choline (*Strecker, 1862*; *Liebreich, 1865*) and acetylcholine (ACh) (*Baeyer, 1867*) were discovered decades before their pharmacological effects were found around the turn of 20th century (*Hunt and Taveau, 1906*; *Mott and Halliburton, 1899*; *Hunt, 1900*). Henry Dale and colleagues realized similarities of ACh and parasympathetic stimulation (*Dale, 1914*; *Ewins, 1914*), but it was not until 1929 when ACh was detected in the body (*Dale and Dudley, 1929*) and 1934 when ACh was proven a neurotransmitter in the peripheral nervous system (PNS) (*Feldberg and Gaddum, 1934*; *Brown and Feldberg, 1936*; *Dale et al., 1936*). It took nearly 100 y from the finding of the effects of supradrenal gland damage (*Addison, 1855*) or removal (*Schäfer, 1908*), the observation of an activity in the supradrenal gland (*Oliver and Schäfer, 1895*), the isolation of an inactive derivative (*Abel, 1898*; *Abel and Crawford, 1897*; *Abel, 1899*; *Abel, 1901*), and the successful isolation of adrenaline (*Takamine, 1901*; *Takamine, 1902*; *Aldrich, 1901*), the notice of similarities between adrenaline and sympathetic stimulation (*Elliot, 1904*; *Elliott, 1905*; *Langley, 1901a*; *Barger and Dale, 1910*; *Cannon et al., 1933*), to the mid-1940s when Ulf von Euler proved that noradrenaline (NA) was the neurotransmitter of the sympathetic nerves (*von Euler, 1946*; *von Euler, 1948*; *von Euler, 1956*). While it is not easy to establish a molecule as a neurotransmitter in the PNS, it is even harder to establish a central nervous system (CNS) neurotransmitter. Three decades elapsed between the time when ACh was proven to be a PNS neurotransmitter and the time when it was established as a CNS neurotransmitter (*Mitchell, 1963*; *Collier and Mitchell, 1966*; *Collier and Mitchell, 1967*) and two decades between NA as a peripheral transmitter and a central transmitter (*Carlsson et al., 1962*; *Björklund et al., 1968*).

If a neurotransmitter acts only in the CNS, but not in the PNS, it is much more difficult to discover or to prove. Most neurotransmitters were discovered for their effects on peripheral tissues, with muscle contraction or relaxation as a major readout. Glutamate (Glu) (*Robbins, 1958*; *Curtis et al., 1959*; *Curtis and Watkins, 1960*; *Curtis and Watkins, 1963*) and gamma-aminobutyric acid (GABA) (*Curtis and Watkins, 1960*; *Roberts and Frankel, 1950*; *Udenfriend, 1950*; *Florey, 1954*; *Florey and McLennan, 1955*) were discovered partly because of their peripheral effects and partly because of their effects on spinal neurons. There is no reason for central neurotransmitters to also act peripherally, but relatively little efforts have been reported to find small-molecule neurotransmitters acting only on CNS neurons with no peripheral bioassays available. Premature assumptions and technical difficulties are among the major reasons why the hunt for neurotransmitters has not been a highly active area of research over the last three decades.

Are there more neurotransmitters and how can they be discovered? Classic neurotransmitters are stored in synaptic vesicles (SVs) (*Fatt and Katz, 1950*; *Fatt and Katz, 1952*; *Del Castillo and Katz, 1954*; *Robertson, 1953*; *Palade and Palay, 1954*; *Palay, 1954*; *De Robertis and Bennett, 1954*; *De Robertis and Bennett, 1955*; *del Castillo and Katz, 1956*). They are released upon electric stimulation before being degraded enzymatically or taken up into the presynaptic terminal by cytoplasmic transporters and into SVs by vesicular transporters (*Axelrod et al., 1959*; *Radian and Kanner, 1985*; *Radian et al., 1986*; *Guastella et al., 1990*; *Pacholczyk et al., 1991*; *Blakely et al., 1991*). Most of the major textbooks list either three (*Purves et al., 2001*; *Purves et al., 2016*; *Bear et al., 2016*) or four (*Kandel et al., 2013*; *Kandel et al., 2021*; *Siegel and Sapru, 2011*) criteria of a neurotransmitter: presence in presynaptic neurons, release upon stimulation, action on postsynaptic neurons, mechanism of removal. Some molecules commonly accepted as neurotransmitters still do not meet all the criteria listed in different textbooks, but they nonetheless play important functional roles in the CNS and their defects cause human diseases. Over time, different small molecules have been proposed to function as neurotransmitters (e.g., *Björklund et al., 1968*; *Felix and Künzle, 1974*), but none satisfies all the criteria. Robust and reliable detection of the candidate molecule in SVs is often, though not always, the problem (cf. *Chantranupong et al., 2020*).

Beginning in 2011, we have been actively searching for new neurotransmitters in the mammalian brain. We have tried different approaches, including searching for neuroactive substances in the

cerebral spinal fluid (CSF) and following transporters potentially localized in the SVs. One approach that we have now taken to fruition is the purification of the SVs from mouse brains coupled with chemical analysis of their contents. We have found known transmitters such as Glu, GABA, ACh, and 5-hydroxytryptamine (5-HT). But more importantly, we have reproducibly detected creatine (Cr) in SVs.

Cr was discovered in 1832 by Michel-Eugène Chevreul (*Chevreul, 1835*; *Liebig, 1847*) and has long been considered as an energy buffer in the muscle and the brain (*Wyss and Kaddurah-Daouk, 2000*; *Brosnan and Brosnan, 2007*; *Wallimann et al., 2011*). Half of Cr in a mammalian animal is thought to come from diet and the rest from endogenous synthesis (*Braissant et al., 2011*). Most of the Cr is present in the muscle, but it is also present in the brain. Although most of the endogenous Cr is synthesized in the kidney, the pancreas, and the liver (*Wyss and Kaddurah-Daouk, 2000*; *Ohtsuki et al., 2002*), Cr is also synthesized in the brain (*Braissant et al., 2011*; *Braissant et al., 2001*; *Braissant et al., 2007*).

Solute carriers (SLC) contribute to both cytoplasmic and vesicular transporters. With 19 members in humans, family 6 (SLC6) are secondary active transporters relying on electrochemical $Na^+$ or $H^+$ gradients (*Nelson, 1998*; *Chen et al., 2004*; *Höglund et al., 2005*; *Bröer and Gether, 2012*; *Iversen, 2006*; *Bröer, 2006*). SLC6 is also known as the neurotransmitter transporter (NTT) family because some members transport neurotransmitters such as GABA (by SLC6A1 or GABA transporter 1, GAT1; SLC6A13 or GAT2; SLC6A11 or GAT3) (*Guastella et al., 1992*; *Clark et al., 1992*; *Borden et al., 1992*; *Lopez-Corcuera et al., 1992*), NA (by SLC6A2 or NA transporter [NET]) (*Pacholczyk et al., 1991*), dopamine (by SLC6A3 or DAT) (*Giros et al., 1991*; *Kilty et al., 1991*; *Shimada et al., 1991*), 5-HT (by SLC6A4 or serotonin transporter [SERT]) (*Blakely et al., 1991*; *Hoffman et al., 1991*), and glycine (by SLC6A9, or GlyT1; SLC6A5 or GlyT2) (*Guastella et al., 1992*; *Smith et al., 1992*; *Liu et al., 1992*). Cr is transported by SLC6A8 (also known as CrT, CT1, or CRCT) (*Mayser et al., 1992*; *Guimbal and Kilimann, 1993*; *Gonzalez and Uhl, 1994*; *Nash et al., 1994*; *Schloss et al., 1994*; *Sora et al., 1994*; *Barnwell et al., 1995*). In addition to peripheral organs and tissues, SLC6A8 is also expressed in the nervous system where it is mainly in neurons (*Braissant et al., 2001*; *Schloss et al., 1994*; *Happe and Murrin, 1995*; *Saltarelli et al., 1996*; *Mak et al., 2009*). SLC6A8 protein could be found on the plasma membrane of neurons (*Mak et al., 2009*; *Speer et al., 2004*; *Lowe et al., 2015*).

The functional significance of SLC6A8 in the brain is supported by symptoms of humans defective in SLC6A8. Mutations in *SLC6A8* were found in human patients with intellectual disability (ID), delayed language development, epileptic seizures, and autistic-like behaviors (*Salomons et al., 2001*; *Margherita Mancardi et al., 2007*). They are collectedly known as Cr transporter deficiency (CTD), with ID as the hallmark. Particular vulnerability of language development has been observed in some *SLC6A8* mutations which had mild ID but severe language delay (*Battini et al., 2011*). CTD contributes to approximately 1–2.1% of X-linked mental retardation (*Rosenberg et al., 2004*; *Newmeyer et al., 2005*; *Clark et al., 2006*; *Lion-François et al., 2006*; *Arias et al., 2007*; *Puusepp et al., 2010*; *Cheillan et al., 2012*; *van de Kamp et al., 2014*; *Joncquel-Chevalier Curt et al., 2015*). While CTD is highly prevalent in ID males, it is also present in females, with an estimated carrier frequency of 0.024% (*DesRoches et al., 2015*).

*Slc6a8* knockout mice (*Skelton et al., 2011*; *Kurosawa et al., 2012*; *Baroncelli et al., 2016*) showed typical symptoms of human CTD patients with early and progressive impairment in learning and memory. Mice with brain- and neuronal-specific knockout of *Slc6a8* showed deficits in learning and memory without changes in locomotion caused by peripheral involvement of *Slc6a8* (*Udobi et al., 2018*; *Molinaro et al., 2019*). Deletion of *Slc6a8* from dopaminergic neurons in the brain caused hyperactivity (*Abdulla et al., 2020*). These results demonstrate that SLC6A8 is functionally important in neurons.

Cr deficiency syndromes (CDS) are inborn errors of Cr metabolism, which can result from defects in one of the three genes: guanidinoacetate methyltransferase (*GAMT*) (*Stöckler et al., 1994*), arginine-glycine amidinotransferase (*AGAT*) (*Bianchi et al., 2000*; *Item et al., 2001*), and *SLC6A8* (*Salomons et al., 2001*). That they all show brain disorders indicates the functional importance of Cr in the brain (*Stockler-Ipsiroglu et al., 2014*; *Stockler-Ipsiroglu et al., 2015*; *Khan et al., 2016*; *Fons and Campistol, 2016*).

Here we first biochemically purified SVs from the mouse brain and discovered the presence of Cr, as well as classic neurotransmitters Glu and GABA, ACh and 5-HT, in SVs. We then detected calcium

(Ca²⁺)-dependent releases of Cr, Glu, and GABA but not ACh and 5-HT when neurons were depolarized by increased extracellular concentrations of potassium (K⁺). Both the level of Cr in SVs and that of Cr released upon stimulation were decreased significantly when either the gene for *Slc6a8* or the gene for *Agat* were eliminated genetically. When Cr was applied to slices from the neocortex, the activities of pyramidal neurons were inhibited. Furthermore, we confirmed that Cr was taken up by synaptosomes and found that Cr uptake was significantly reduced when the *Slc6a8* gene was deleted. Finally, we found that Cr was transported into SVs. Thus, multidisciplinary studies with biochemistry, genetics, and electrophysiology have suggested that Cr is a new neurotransmitter, though the discovery of a receptor for Cr would prove it.

## Results
### Detection of Cr in SVs from the mouse brain
To search for new neurotransmitters, we tried several approaches. For example, we used Ca²⁺ imaging to detect neuroactive substances in the cerebrospinal fluid (CSF), but it was difficult to rule out existing neurotransmitters and select responses from potentially new neurotransmitters. We also transfected cDNAs for all human SLCs into dissociated cultures of primary neurons from the mouse brain and found that more than 50 out of all SLCs could be localized in SVs. However, when we used CRISPR-Cas9 to tag some of the candidate SLCs in mice, some of them were found to be expressed outside the CNS, indicating that, while ectopic expression of these candidate SLCs could be localized on SVs, the endogenous counterparts were not localized on SVs.

Here, we report our approach using the purification of SVs as the first step (*Figure 1—figure supplement 1A*; *Whittaker et al., 1964*; *Nagy et al., 1976*; *Matthew et al., 1981*; *De Camilli et al., 1983*; *Huttner et al., 1983*; *Burger et al., 1989*; *Burger et al., 1991*; *Maycox et al., 1992*). Synaptophysin (Syp) is a specific marker for SVs (*Jahn et al., 1985*; *Wiedenmann and Franke, 1985*; *Leube et al., 1987*; *Südhof et al., 1987*) and an anti-Syp antibody was used to immunoisolate SVs (*Burger et al., 1989*; *Burger et al., 1991*; *Jahn and Sudhof, 1994*; *Martineau et al., 2013*; *Bradberry et al., 2022*). Visualization by electronic microscopy (EM) (*Figure 1—figure supplement 1B*, left panel) showed that the purified vesicles were homogeneous, with an average diameter of 40.44 ± 0.26 nm (n = 596 particles) (*Figure 1—figure supplement 1B*, right panel), consistent with previous descriptions of SVs (*Ahmed et al., 2013*; *Takamori et al., 2006*).

Immunoblot analysis with 20 markers of subcellular organelles of neurons and 1 marker for glia (*Figure 1—figure supplement 1C*) indicates that our purifications were highly effective, with SV markers detected after purification with the anti-Syp antibody, but not that with the control immunoglobulin G (IgG). SV proteins included Syp (*Jahn et al., 1985*; *Wiedenmann and Franke, 1985*; *Leube et al., 1987*; *Südhof et al., 1987*), synaptotagmin (Syt1) (*Geppert et al., 1994*), synatobrevin2 (Syb2) (*Link et al., 1992*; *Schiavo et al., 1992*), SV2A (*Bajjalieh et al., 1992*), H⁺-ATPase (*Cidon and Sihra, 1989*), and vesicular neurotransmitter transporters for glutamate (VGLUT1, VGLUT2) (*Takamori et al., 2000*) and GABA (VGAT) (*McIntire et al., 1997*). Immunoisolation by the anti-Syp antibody did not bring down markers for the synaptic membrane (with SNAP23 as a marker) (*Kunii et al., 2021*; *Ravichandran et al., 1996*; *Suh et al., 2010*), postsynaptic components (with PSD95 and GluN1 as markers) (*Woods and Bryant, 1991*; *Cho et al., 1992*), the Golgi apparatus (with GM130 and Golgin 97 as markers) (*Nakamura et al., 1995*; *Griffith et al., 1997*), early endosome (with early endosome-associated 1 [EEA1] as a marker) (*Mu et al., 1995*), the lysosome (with LC3B and cathepsinB as markers), the cytoplasma (with glyceraldehyde-3-phophate dehydrogenase [GAPDH] as a marker), mitochondria (with voltage-dependent anion channel [VDAC] as a marker), cytoplasmic membrane (with calcium voltage-gated channel subunit alpha 1 [CACNA1A] as a marker), axonal membrane (with glucose transporter type 4 [GluT4] as a marker), and glia membrane (with myelin basic protein [MBP] as a marker). These results indicated that the SVs we obtained were of high integrity and purity.

To detect and quantify small molecules as candidate transmitters present in the purified SVs, capillary electrophoresis-mass spectrometry (CE-MS) was optimized and utilized (*Figure 1A*, *Figure 1—figure supplement 1A*; *Martineau et al., 2013*; *Tie et al., 2012*). We found that the levels of classical neurotransmitters such as Glu, GABA, ACh, and 5-HT were significantly higher in SVs pulled down by the anti-Syp antibody than those in lysates pulled down by the control IgG (*Figure 1A–E*). Consistent with previous reports (*Burger et al., 1991*; *Martineau et al., 2013*), significant enrichment of

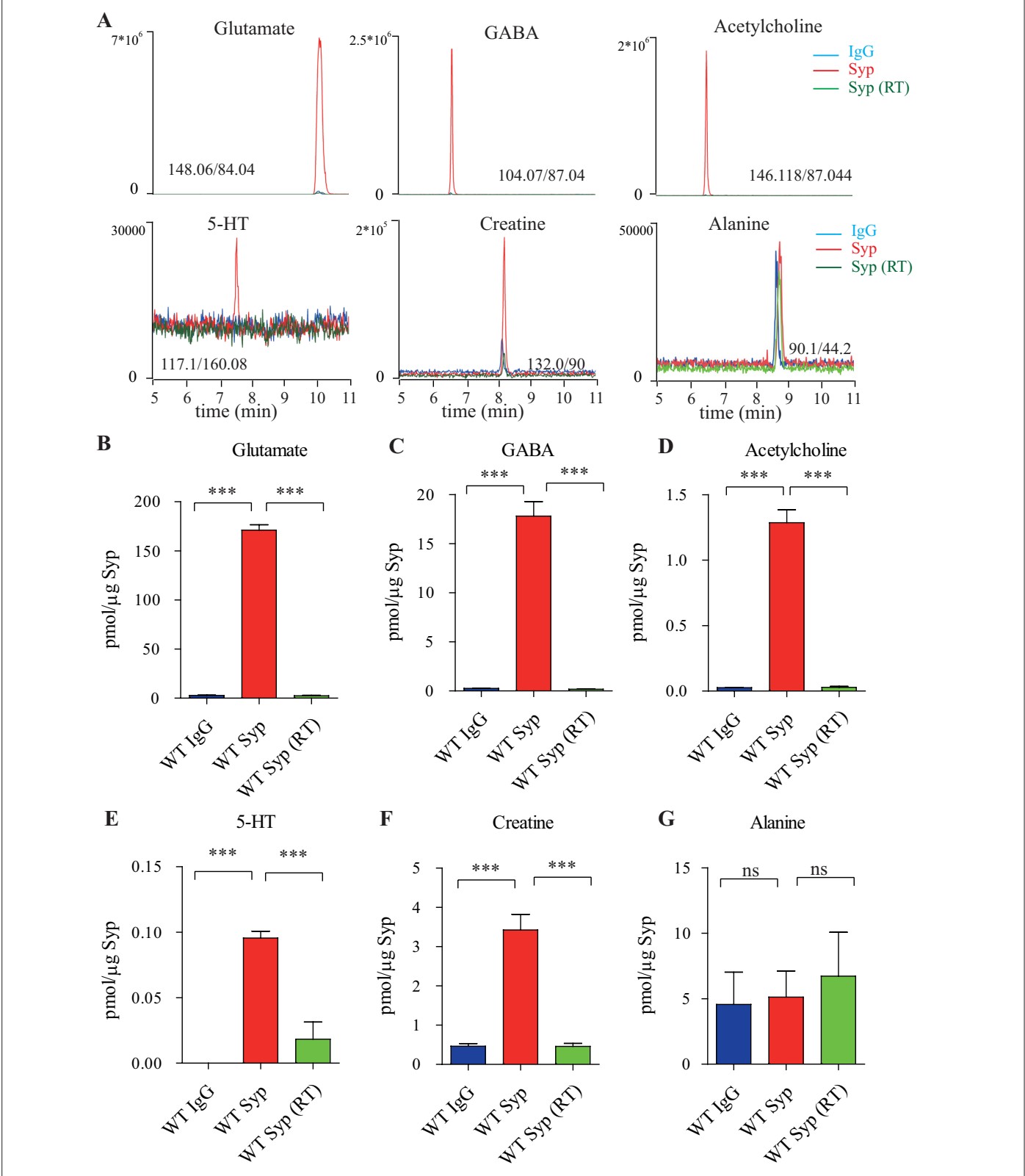

**Figure 1.** Presence of creatine (Cr) in synaptic vesicles (SVs) from the mouse brain. (**A**) Representative raw traces from capillary electrophoresis-mass spectrometry (CE-MS) of indicated molecules from samples immunoisolated by the control immunoglobulin G (IgG) (blue) at 0–2°C, the monoclonal anti-synaptophysin (anti-Syp) antibody at 0–2°C (red), and the anti-Syp antibody at room temperature (RT, green). Q1/Q3 for identifying targets are indicated. (**B–G**) Quantification of the amounts of indicated molecules. The amount of a molecule was divided by the amount of the anti-Syp antibody

*Figure 1 continued on next page*

*Figure 1 continued*

bound to magnetic beads. Note glutamate (Glu) (**B**), gamma-aminobutyric acid (GABA)(C), acetylcholine (ACh) (**D**), 5-hydroxytryptamine (5-HT) (**E**), Cr (**F**), but not alanine (**G**) was higher in SVs pulled down by the anti-Syp antibody at 0–2°C than those pulled down by the IgG control or those pulled down at the RT. n = 10 (**B–E, G**) or 14 (**F**) samples per group, *** p<0.001, ns, not significant. One-way ANOVA with Tukey's correction.

The online version of this article includes the following source data and figure supplement(s) for figure 1:

**Source data 1.** Data for *Figure 1B-G*.

**Figure supplement 1.** Validation of synaptic vesicle (SV) purification from the mouse brain.

**Figure supplement 1—source data 1.** Particle diameter of SVs for *Figure 1—figure supplement 1B*.

**Figure supplement 1—source data 2.** Original western blot files for *Figure 1—figure supplement 1C*.

**Figure supplement 1—source data 3.** Labelled western blot files for *Figure 1—figure supplement 1C*.

**Figure supplement 2.** Effects of pharmacological inhibitors on synaptic vesicle (SV) contents.

**Figure supplement 2—source data 1.** Data for *Figure 1—figure supplement 2A–E*.

neurotransmitters was observed only from SVs immunoisolated at near 0°C, but not at the room temperature (RT) (*Figure 1A–E*). By contrast, another small molecule, alanine (*Figure 1G*), was not elevated in SVs compared to the control.

The amount of Glu was 171.1 ± 5.4 pmol/µg anti-Syp antibody (n = 10, *Figure 1B*), approximately 10 times that of GABA (n = 10,17.81 ± 1.47 pmol/µg anti-Syp antibody, *Figure 1C*). The amount of ACh was 1.29 ± 0.10 pmol/µg anti-Syp antibody (n = 10, *Figure 1D*), approximately 0.072 that of GABA. The amount of 5-HT was 0.096 ± 0.017 pmol/µg anti-Syp antibody (n = 10, *Figure 1E*). Thus, our purification and detection methods were highly reliable and sensitive enough to detect established neurotransmitters.

Under the same conditions, we also detected Cr in SVs (n = 14, *Figure 1A, F*). Amount of Cr in the SVs was found to be 3.43 ± 0.40 pmol/µg anti-Syp antibody (*Figure 1F*), which was approximately 2% of Glu, 19% of GABA, 266% of ACh, and 3573% of 5-HT. It is unlikely that these could be attributable to different levels for different neurotransmitters in each SV, but more likely attributable to the relative abundance of SVs containing different neurotransmitters. Also, 85–90% neurons in the mouse brain were glutamatergic while 10–15% were GABAergic (*Meyer et al., 2011*; *Olbrich and Braak, 1985*; *Tremblay et al., 2016*), which can explain our detection of Glu as approximately 10 times that of GABA (*Figure 1B, C*). Similarly, cholinergic neurons ($5.67 \times 10^5$) (*Li et al., 2018*) represented 0.81% of total number of neurons (approximately 70 million) in the mouse brain (*Herculano-Houzel et al., 2006*), serotonergic neurons (approximately 26,000) for 0.037% of total neurons (*Herculano-Houzel et al., 2006*; *Ishimura et al., 1988*). Assuming that the content of each neurotransmitter in a single SV is similar, extrapolation from the above data would suggest that approximately 1.3–2.15% of neurons in the mouse brain are creatinergic.

To distinguish whether small molecules co-purified with SVs were in the SVs (*Burger et al., 1989*; *Burger et al., 1991*), or that they were just associated with the outside of SVs (*Takamori et al., 2006*), we tested the dependence of the presence of these molecules in the SVs on temperature and on the electrochemical gradient of $H^+$. Cr was significantly reduced in SVs purified at RT compared to that immunoisolated at near 0°C (*Figure 1F*), supporting the presence of Cr inside, instead of outside, SVs.

Classical neurotransmitters are stored in SVs with an acidic environment inside (pH of 5.6–6.4) (*Egashira et al., 2016*; *Mani and Ryan, 2009*; *Egashira et al., 2015*). To further verify the storage of Cr in SVs and examine the role of $H^+$ electrochemical gradient, we applied pharmacological inhibitors during purification (*Chantranupong et al., 2020*; *Qian et al., 2021*). The proton ionophore FCCP (carbonyl cyanide-4-(tri-fluoromethoxy) phenylhydrazone) was used to dissipate $H^+$ electrochemical gradient (*Qian et al., 2021*; *Schenck et al., 2009*). FCCP significantly reduced the amount of Cr as well as classical neurotransmitters in SVs (*Figure 1—figure supplement 2A–E*). The extent of FCCP-induced reduction was correlated with the value of pKa or PI (isoelectric point) for different molecule: 5-HT (with pKa predicted to 10 and 9.31, *Figure 1—figure supplement 2E*) > Cr (PI of ~7.94, *Figure 1—figure supplement 2A*) > GABA (PI of 7.33, *Figure 1—figure supplement 2C*) > Glu (PI of 3.22, *Figure 1—figure supplement 2B*). Nigericin, a $K^+/H^+$ exchanger which dissipates ΔpH (*Qian et al., 2021*; *Schenck et al., 2009*), also reduced the amount of Cr and classical neurotransmitters in SVs (*Figure 1—figure supplement 2A–E*). Furthermore, in the presence of FCCP or nigericin, SV Cr was reduced to a level comparable to that pulled down by control IgG (*Figure 1—figure supplement*

*2A*), demonstrating the storage of Cr in SVs was dependent on H$^+$ gradient. As a control, the non-neurotransmitter molecule alanine in SVs was not changed by the inhibitors (*Figure 1—figure supplement 2F*).

## Reduction of SV Cr in mouse mutants lacking *Slc6a8*

*SLC6A8*, located on the X chromosome, encodes a transporter for Cr and its loss-of-function (LOF) mutations caused behavioral deficits in humans (*Salomons et al., 2001*; *Margherita Mancardi et al., 2007*) and mice (*Skelton et al., 2011*; *Kurosawa et al., 2012*; *Baroncelli et al., 2016*; *Udobi et al., 2018*; *Molinaro et al., 2019*; *Abdulla et al., 2020*). To investigate whether SLC6A8 affects Cr in SVs, we generated *Slc6a8* knockout (KO) mice. Exon 1 of the *Slc6a8* gene was partially replaced with CreERT2-WPRE-polyA by CRISPR/Cas9 (*Figure 2A*). Examination by reverse polymerase chain reaction (RT-PCR) (*Figure 2—figure supplement 1A, B*) and quantitative real-time reverse PCR (qPCR, *Figure 2—figure supplement 1C, D*) showed that *Slc6a8* mRNA was not detected in either male or female mutants, and significantly reduced in female heterozygous (*Slc6a8$^{+/-}$*). Consistent with previous reports, the body weights of *Slc6a8* KO mice were reduced (*Figure 2—figure supplement 2B, D*; *Skelton et al., 2011*; *Duran-Trio et al., 2021*; *Stockebrand et al., 2018*). Brain weight was not significantly different between *Slc6a8* KO mice and WT mice (*Figure 2—figure supplement 2A, C*).

When we examined the contents of SVs isolated by the anti-Syp antibody vs the control IgG, significant reduction was only observed for Cr, but not classical neurotransmitters (*Figure 2B–H*, *Figure 2—figure supplement 3A–E*). While Cr pulled down by IgG was not significantly different between *Slc6a8$^{-/Y}$* and *Slc6a8$^{+/Y}$* mice, SV Cr purified by the anti-Syp antibody from *Slc6a8$^{-/Y}$* was reduced to approximately 1/3 that of the WT (*Slc6a8$^{+/Y}$*) littermates (n = 14, *Figure 2B and C*). Compared to the IgG control, Cr in SVs was enriched in WT mice, but not in *Slc6a8$^{-/Y}$* mice (*Figure 2B and C*). In both *Slc6a8$^{-/Y}$* and *Slc6a8$^{+/Y}$* mice, classical neurotransmitters in SVs were all enriched as compared to IgG controls (*Figure 2D–G*, *Figure 2—figure supplement 3A–D*). The amounts of Glu (*Figure 2D*, *Figure 2—figure supplement 3A*), GABA (*Figure 2E*, *Figure 2—figure supplement 3B*), ACh (*Figure 2F*, *Figure 2—figure supplement 3C*), and 5-HT (*Figure 2G*, *Figure 2—figure supplement 3D*) in SVs were not different between *Slc6a8$^{-/Y}$* and *Slc6a8$^{+/Y}$* mice. Molecules not enriched in SVs from WT mice, such as alanine, were also unaffected by *Slc6a8* KO (*Figure 2H*, *Figure 2—figure supplement 3E*).

It is unlikely that the specific reduction of Cr in SVs from *Slc6a8* KO mice was due to technical artifacts. First, the possibility of less SVs obtained from *Slc6a8* KO mice was precluded by immunoblot analysis, as assessed by SV markers Syp, Syt, and H$^+$-ATPase (*Figure 2—figure supplement 4A*). Second, data collected by high-resolution MS (Q Exactive HF-X, Thermo Scientific, Waltham, MA) also revealed selective decrease of SV Cr (m/z = 132.0576) from *Slc6a8* KO mice (n = 8, *Figure 2—figure supplement 4B–G*), as quantified by the peak area. Peak areas for Glu (n = 8, *Figure 2—figure supplement 4B*, m/z = 148.0604), GABA (n = 8, *Figure 2—figure supplement 4C*, m/z = 104.0712), ACh (n = 8, *Figure 2—figure supplement 4D*, m/z = 146.1178), and alanine (n = 8, Ala, *Figure 2—figure supplement 4E*, m/z = 90.055) were not significantly different between SVs immmunoisolated with the anti-Syp antibody and control IgG from WT and *Slc6a8* KO mice. However, peak areas (*Figure 2—figure supplement 4G*) and amplitude of Cr (n = 8, *Figure 2—figure supplement 4F*) signal were significantly increased in SVs from WT mice (anti-Syp antibody vs IgG), but not that from *Slc6a8* KO mice.

## Reduction of SV Cr in mouse mutants lacking *Agat*

AGAT is the enzyme catalyzing the first step in Cr synthesis (*Braissant et al., 2001*; *Guthmiller et al., 1994*) and its absence also led to Cr deficiency in the human brain and mental retardation (*Bianchi et al., 2000*; *Item et al., 2001*).

To investigate the requirement of AGAT for SV Cr, we utilized *Agat* 'knockout-first' mice (Figure 6A; *Skarnes et al., 2011*). The targeting cassette containing Frt (Flip recombination sites)-flanked EnS2A, an IRES::lacZ trapping cassette, and a floxed *neo* cassette were inserted downstream of exon 2 to interfere with normal splicing of *Agat* pre-mRNA. Examination by RT-PCR (*Figure 3—figure supplement 1A*) and quantitative RT-PCR (*Figure 3—figure supplement 1B*) showed reduction of *Agat* mRNA in *Agat$^{+/-}$* and absence of *Agat* mRNA in *Agat$^{-/-}$* mice. Body weight (*Figure 3—figure*

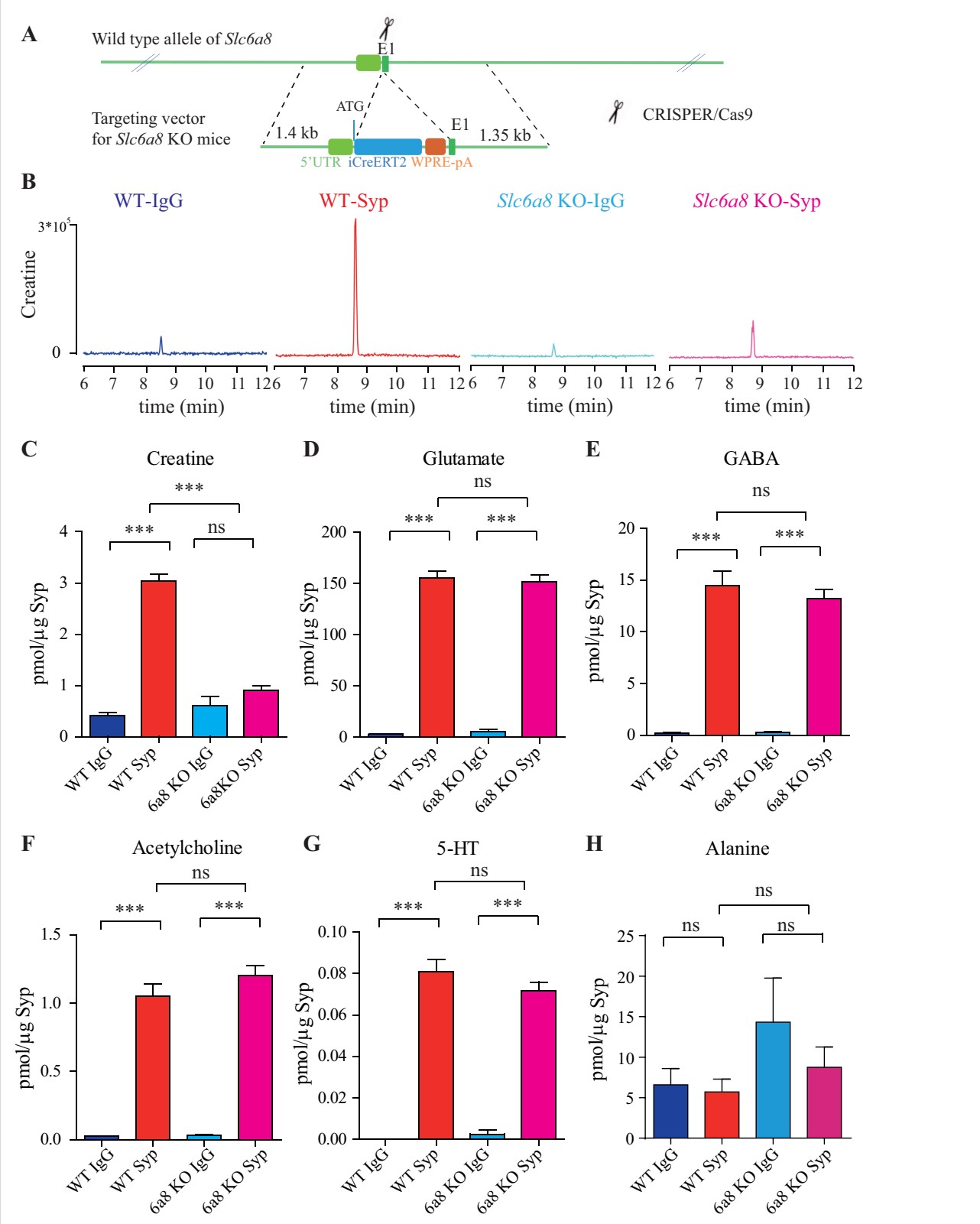

**Figure 2.** *Slc6a8* and creatine (Cr) in synaptic vesicles (SVs). (**A**) A schematic illustration of the strategy for generating *Slc6a8* knockouts using CRISPR/Cas9. An iCreERT2-WPRE-pA cassette (~3.5 kb) was inserted immediately downstream of ATG in the *Slc6a8* gene, substituting bp 4 to bp 51 in exon 1 (E1). (**B**) Representative raw traces of Cr immunoisolated by control immunoglobulin G (IgG) from WT mice (blue), the anti-synaptophysin (anti-Syp) antibody from WT mice (red), IgG from *Slc6a8* KO mice (blue), and the anti-Syp antibody from *Slc6a8* KO mice (red). (**C–H**) Quantification of indicated

*Figure 2 continued on next page*

*Figure 2 continued*

molecules. Note the selective reduction of Cr in SVs from *Slc6a8* KO mice. n = 14 samples per group. ***p<0.001, ns, not significant. One-way ANOVA with Tukey's correction.

The online version of this article includes the following source data and figure supplement(s) for figure 2:

**Source data 1.** Data for *Figure 2C–H*.

**Figure supplement 1.** Validation of SLC6A8 knockout mice.

**Figure supplement 1—source data 1.** Original gels for *Figure 2—figure supplement 1A and B*.

**Figure supplement 1—source data 2.** Labelled gels for *Figure 2—figure supplement 1A and B*.

**Figure supplement 1—source data 3.** Data for *Figure 2—figure supplement 1C and D*.

**Figure supplement 2.** Brain and body weights of *Slc6a8* knockout mice.

**Figure supplement 2—source data 1.** Data for *Figure 2—figure supplement 2A–D*.

**Figure supplement 3.** Representative capillary electrophoresis-mass spectrometry (CE-MS) data of molecules.

**Figure supplement 4.** Proteins and small molecules detected in synaptic vesicles (SVs) from WT and *Slc6a8* KO mice.

**Figure supplement 4—source data 1.** Original western blot files for *Figure 2—figure supplement 4A*.

**Figure supplement 4—source data 2.** Labelled western blot files for *Figure 2—figure supplement 4A*.

**Figure supplement 4—source data 3.** Data for *Figure 2—figure supplement 4B–E, and G*.

supplement 2B), but not brain weight (*Figure 3—figure supplement 2A*), of *Agat*[-/-] mice was lower than both *Agat*[+/+] and *Agat*[+/-] mice, which were similar to *Slc6a8* KO mice.

Immunoblot analysis showed SVs purified from the brains were not significantly different among *Agat*[+/+], *Agat*[+/-], and *Agat*[-/-] mice (*Figure 3—figure supplement 3A–D*), as supported by quantitative analysis of Syp, Syt, and H[+]-ATPase (n = 20, with two repeats for 10 samples).

We analyzed small molecules present in SVs from *Agat*[+/+], *Agat*[+/-], and *Agat*[-/-] mice. Cr was significantly enriched in SVs from all three genotypes compared to the IgG control (+/-). However, the level of Cr from *Agat*[-/-] mice was significantly lower than those from *Agat*[+/+] and *Agat*[+/-] (n = 10, *Figure 3A and B*).

Glu (*Figure 3C*), ACh (*Figure 3E*), and 5-HT (*Figure 3F*) were all enriched in SVs (compared to IgG controls) and not significantly different among *Agat*[+/+], *Agat*[+/-], and *Agat*[-/-] mice. GABA in SVs from *Agat*[-/-] mice was also decreased from *Agat*[+/+] mice by 30%, to an extent less than that of Cr (78.4%). Alanine was not different among three genotypes of mice (n = 6, *Figure 3G*). Thus, Cr and GABA, but not other neurotransmitters, in SVs were reduced in *Agat* KO mice.

## Pattern of SLC6A8 expression indicated by knockin mice

We generated *Slc6a8*[HA] knockin mice by CRISPR/Cas9. Three repeats of the hemagglutinin (HA) tag (*Kolodziej and Young, 1991*), the T2A sequence (*Ryan et al., 1991*; *Ahier and Jarriault, 2014*; *Daniels et al., 2014*), and CreERT2 (*Sauer and Henderson, 1989*; *Gu et al., 1993*; *Feil et al., 1997*; *Indra et al., 1999*) were inserted in-frame at the C terminus of the SLC6A8 protein (*Figure 4A*).

To examine the expression pattern of SLC6A8, we performed immunocytochemistry with an antibody against the HA epitope in *Slc6a8*[HA] and WT mice. *Slc6a8*[HA] mice showed positive signals in the olfactory bulb (*Figure 4B*), the piriform cortex (*Figure 4C–F*), the somatosensory cortex (*Figure 4F and G*), the ventral posterior thalamus (*Figure 4H*), the interpeduncular nucleus (*Figure 4I*), and the pontine nuclei (*Figure 4J*). In addition, moderate levels of immunoreactivity were observed in the motor cortex (*Figure 4D–H*), the medial habenular nucleus (*Figure 4H*), the hippocampus (*Figure 4H*), and the cerebellum (*Figure 4K*). These results were consistent with previous reports (*Mak et al., 2009*; *Lowe et al., 2015*). WT mice were negative for anti-HA antibody staining (*Figure 4L and M*).

## Ca[2+]-dependent release of Cr upon stimulation

Classical neurotransmitters are released from the SVs into the synaptic cleft in a Ca[2+]-dependent manner after stimulation. For example, high extracellular potassium (K[+]) stimulated Ca[2+]-dependent release of Glu, GABA, and other neurotransmitters in brain slices (*Hamberger et al., 1979b*; *Hamberger et al., 1979a*; *McBride et al., 1983*; *Nadler et al., 1977*; *Keith et al., 1993*).

Thus, 300-µm-thick coronal slices of the mouse brain within 1–2 mm posterior to the bregma were used because the cortex, the thalamus, the habenular nucleus, and the hippocampus were positive

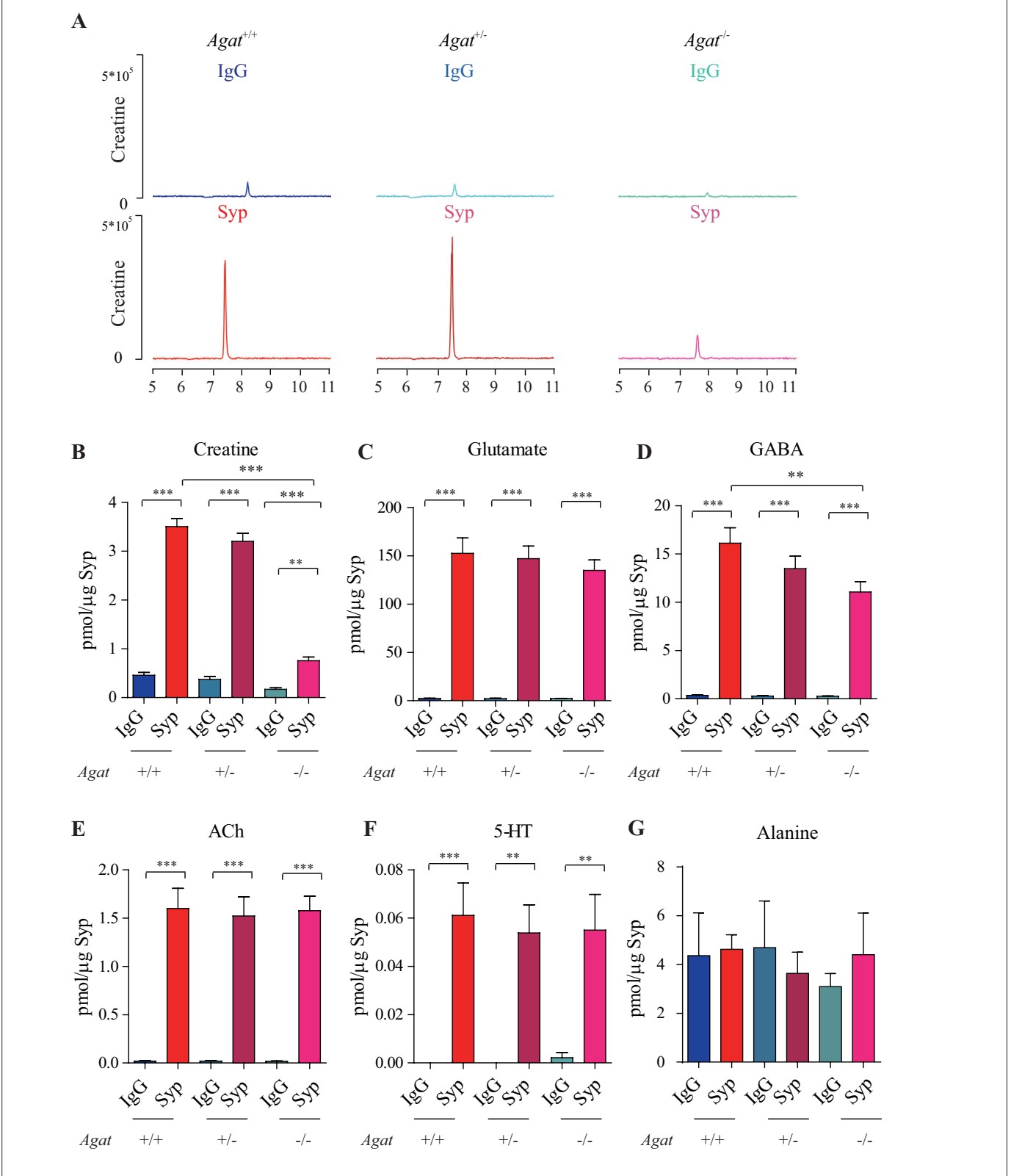

**Figure 3.** Reduction of synaptic vesicle (SV) creatine (Cr) in *Agat* knockout mice. (**A**) Representative raw traces of Cr, pulled down by immunoglobulin G (IgG) or anti-synaptophysin (anti-Syp) from A*gat*<sup></sup>, A*gat* <sup></sup>, and A*gat*<sup></sup> mice. (**B–G**) Quantification of indicated molecules. Cr was significantly decreased in A*gat*<sup></sup> mice compared to Cr in A*gat*<sup></sup> or A*gat* <sup></sup> mice (**B**). Gamma-aminobutyric acid (GABA) was significantly decreased in SVs from A*gat*<sup></sup> mice compared to A*gat*<sup></sup> mice (**D**) but the difference was smaller than that of Cr. Glutamate (Glu) (**C**), acetylcholine (ACh) (**E**), 5-hydroxytryptamine (5-HT)

*Figure 3 continued on next page*

*Figure 3 continued*

(**F**), and alanine was not different among A*gat*$^{+/+}$, A*gat* $^{+/-}$, and A*gat*$^{-/-}$mice. n = 10 samples per group. ***p<0.001, ns, not significant. One-way ANOVA with Tukey's correction.

The online version of this article includes the following source data and figure supplement(s) for figure 3:

**Source data 1.** Data for *Figure 3B–G*.

**Figure supplement 1.** Validation *Agat*-KO first mice.

**Figure supplement 1—source data 1.** Original gels for *Figure 3—figure supplement 1A*.

**Figure supplement 1—source data 2.** Labelled gels for *Figure 3—figure supplement 1A*.

**Figure supplement 1—source data 3.** Data for *Figure 3—figure supplement 1B*.

**Figure supplement 2.** Brain and body weights of *Agat*$^{+/+}$, *Agat*$^{+/-}$, and *Agat*$^{-/-}$ mice.

**Figure supplement 2—source data 1.** Data for *Figure 3—figure supplement 2A and B*.

**Figure supplement 3.** Similar amounts of synaptic vesicles (SVs) pulled down from *Agat*$^{+/+}$, *Agat*$^{+/-}$, and *Agat*$^{-/-}$ mice.

**Figure supplement 3—source data 1.** Original western blot files for *Figure 3—figure supplement 3A*.

**Figure supplement 3—source data 2.** Labelled western blot files for *Figure 3—figure supplement 3A*.

**Figure supplement 3—source data 3.** Data for *Figure 3—figure supplement 3B–D*.

for SLC6A8 (cf., *Figure 4H*). We monitored the effect of K$^+$ stimulation by recording neurons in the slices. Immediately after K$^+$ stimulation, pyramidal neurons in the CA1 region of the hippocampus were depolarized, firing a train of action potentials and reaching a large depolarization plateau in less than 1 min (*Figure 5A*). K$^+$-induced depolarization persisted for several minutes before returning to the baseline and being washed within 10 min. Thus, superfusates in 1 min fraction at the time points of 1.5 min before (control) and after K$^+$ stimulation, and 10 min after the wash were collected (*Figure 5A*), and the metabolites in the superfusates were analyzed by CE-MS.

In the presence of Ca$^{2+}$, depolarization with elevated extracellular K$^+$ led to robust release of Glu and GABA in slices from WT (*Slc6a8*$^{+/Y}$) mice (n = 7 per group, *Figure 5B and C*). After 10 min wash, levels of Glu and GABA returned to the baseline (*Figure 5B and C*). In the presence of Ca$^{2+}$, depolarization with elevated K$^+$ led to robust release of Cr. Extracellular Cr returned to the baseline after 10 min wash (*Figure 5D*). For quantification, the stimulated releases of metabolites were calculated by subtracting the basal levels from the total releases in response to K$^+$ stimulation. In the presence of Ca$^{2+}$, K$^+$ stimulation induced the efflux of Glu, GABA, and Cr at 0.46, 0.33, and 0.086 nmol/min, respectively (n = 7 per group) (*Figure 5B–D*). From the detection limits of ACh and 5-HT in our system, we inferred that the efflux rate for ACh was lower than 0.001 nmol/min and that for 5-HT lower than 0.003 nmol/min. The efflux rate for Cr in brain slices is lower than those of Glu and GABA, but higher than those for ACh and 5-HT.

Ca$^{2+}$ dependence of transmitter release was examined by comparing responses to ACSF without Ca$^{2+}$ or elevated K$^+$ (supplemented with 1 mM EGTA), elevated extracellular K$^+$ in the absence of Ca$^{2+}$ (supplemented with 1 mM EGTA), or K$^+$ in the presence of 2.5 mM Ca$^{2+}$ (*Figure 5E–G*, n = 5 per group). In the absence of Ca$^{2+}$, elevated K$^+$ stimulated the release of a small but significant amount of Glu and GABA, with efflux rates at 0.056 nmol/min and 0.066, respectively (*Figure 5E and F*). In the presence of 2.5 mM Ca$^{2+}$, elevated K$^+$ further augmented the release of Glu and GABA by 5–6 times, confirming previously reported Ca$^{2+}$-dependent release of neurotransmitters in response to depolarization (*Hamberger et al., 1979b*; *McBride et al., 1983*).

Cr was also released both in a Ca$^{2+}$-dependent and a Ca$^{2+}$-independent manner (*Figure 5G*). More Cr was released in response to K$^+$ stimulation in the presence of 2.5 mM Ca$^{2+}$ than that in the absence of Ca$^{2+}$. These results demonstrate Ca$^{2+}$-dependent release of Cr upon stimulation.

## Reduced Cr release in *Slc6a8* and *Agat* mutant mice

We examined whether *Slc6a8* KO affected K$^+$-induced release of Cr. While Glu and GABA were released in slices from *Slc6a8* KO (*Slc6a8*$^{+/Y}$) mice at levels not significantly different from those of WT mice (*Figure 5B and C*), release of Cr in response to K$^+$ stimulation was significantly reduced in *Slc6a8*$^{-/Y}$ mice compared to *Slc6a8*$^{+/Y}$ mice (*Figure 5D*). The basal level of Cr in *Slc6a8* KO mice was lower than that of WT mice. In addition, K$^+$ stimulation-induced release of Cr persisted to some extent

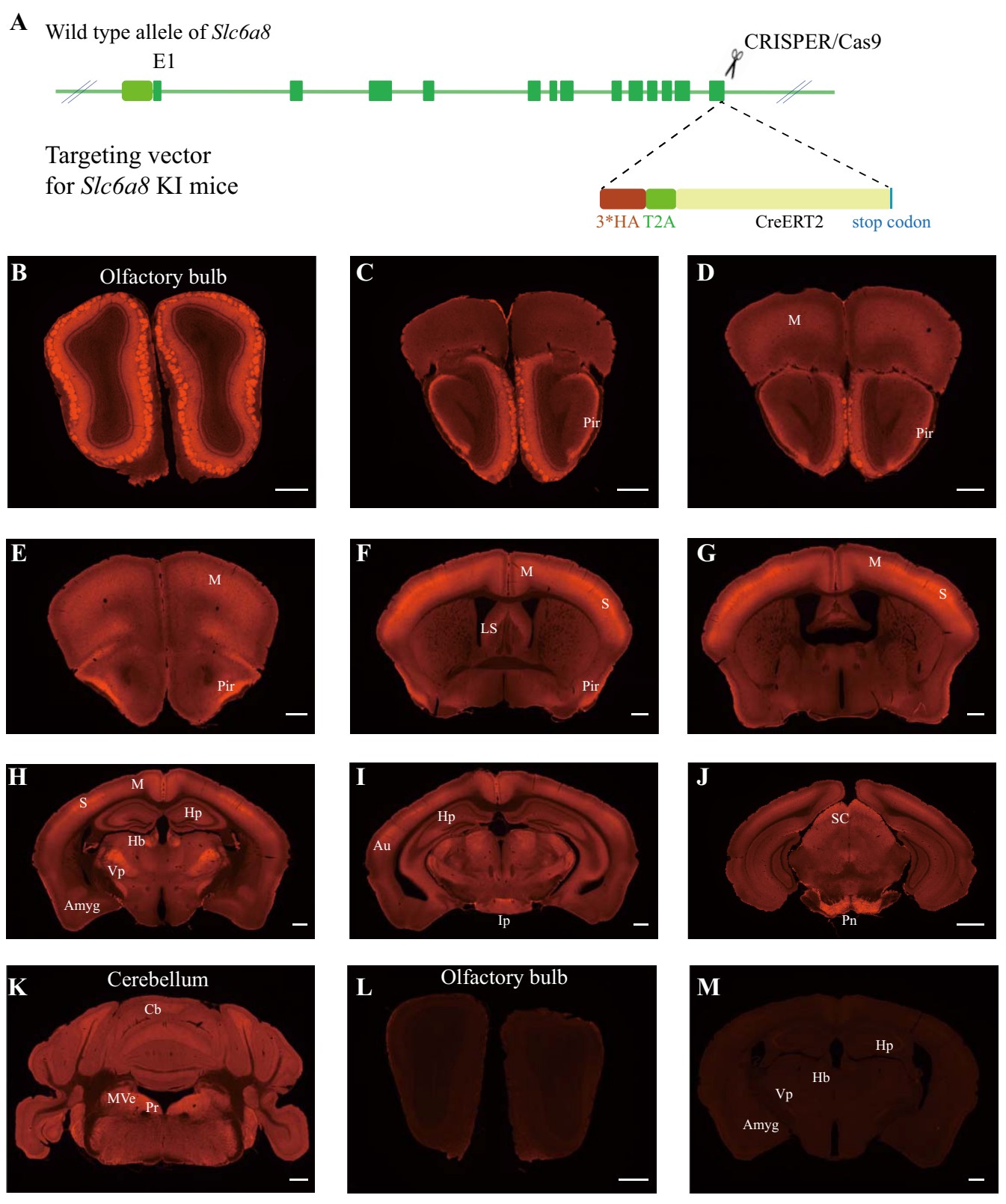

**Figure 4.** Expression pattern of SLC6A8. (**A**) A diagram illustrating the generation of *Slc6a8*^HA KI mice. 3*HA, T2A, and CreERT2 were inserted in-frame, before the stop codon, to the C terminus of SLC6A8 protein. (**B–K**) Low-magnification photomicrographs of coronal sections immunohistochemically labeled with the anti-HA antibody in *Slc6a8*^HA mice. (**I, M**) immunostaining with the anti-HA antibody in control WT mice. Pir, piriform cortex; M, motor cortex; LS, lateral septum; Hp, hippocampus; Hb, habenular nucleus; Vp, ventral posterior nucleus of thalamus; Auauditory cortex; Amyg, amygdala; Ip, interpeduncular nucleus; Pn, pontine nucleus; Cb, cerebellum; Pr, prepositus; SC, superior colliculus; MVe, medial vestibular nucleus. Scale bar: 500 μm.

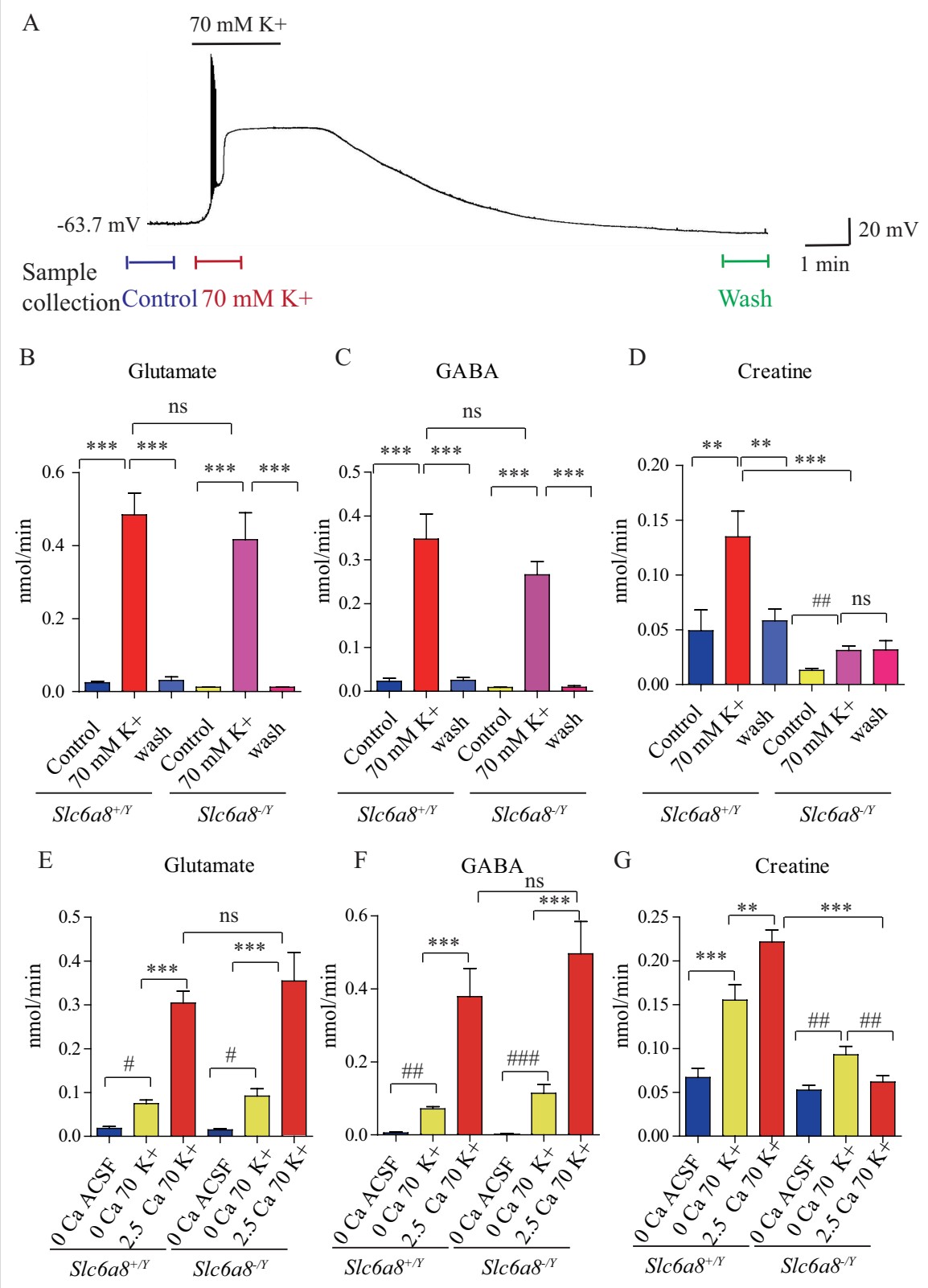

**Figure 5.** Creatine (Cr) release in brain slices from WT and *Slc6a8* knockout mice. (**A**) Neuronal depolarization induced by 70 mM K+ and time points for collecting the release sample. 'Control' samples were collected 1.5 min to 0.5 min before K+ stimulation, '70 mM K+' artificial cerebrospinal fluid (ACSF) samples were collected during 70 mM K+ stimulation, and 'wash' samples were collected 10 min after washout with ACSF. Efflux of glutamate (Glu) (**B**), gamma-aminobutyric acid (GABA) (**C**), or Cr (**D**) from WT or *Slc6a8* KO male mice (n = 7 samples per group). Note that a small amount of Cr

*Figure 5 continued on next page*

*Figure 5 continued*

released in *Slc6a8* KO mice did not return to the baseline after 10 min of washing. (**E–G**) $Ca^{2+}$-dependent release of Glu, GABA, and Cr in WT and *Slc6a8* KO mice (n = 5 samples per group). ***p<0.001, ns, not significant. One-way ANOVA with Tukey's correction. #p<0.05, ##p<0.01, paired *t*-test.

The online version of this article includes the following source data for figure 5:

**Source data 1.** Data for *Figure 5B–G*.

even after 10 min of washout (*Figure 5D*), possibly due to the inability of presynaptic terminals in *Slc6a8* KO mice to reuptake Cr in the synaptic cleft (Figure 8).

Experiments with slices from brains of *Slc6a8* KO (*Slc6a8$^{-/Y}$*) mice showed that $Ca^{2+}$-dependent release of either Glu or GABA was not affected by the genotype of *Slc6a8* (*Figure 5E and F*). By contrast, $Ca^{2+}$-dependent release of Cr was abolished in *Slc6a8$^{-/Y}$* slices. Interestingly, $Ca^{2+}$-independent release of Cr was reduced by a third, but did not reach statistical significance, in *Slc6a8$^{-/Y}$* slices. In the absence of $Ca^{2+}$, the basal level of Cr was not changed in *Slc6a8* KO mice. Taken together, these results indicate that there is $Ca^{2+}$-dependent release of Cr upon stimulation and that SLC6A8 is required specifically for $Ca^{2+}$-dependent release of Cr, but not for $Ca^{2+}$-dependent release of other neurotransmitters such as Glu and GABA, or for $Ca^{2+}$-independent release of Cr.

Knockout of *Agat* (*Figure 6A*) selectively reduced $K^+$ evoked release of Cr, but not those of Glu or GABA (n = 5 per group, *Figure 6B–D*). Although $K^+$ stimulation still elicited Cr release from brain

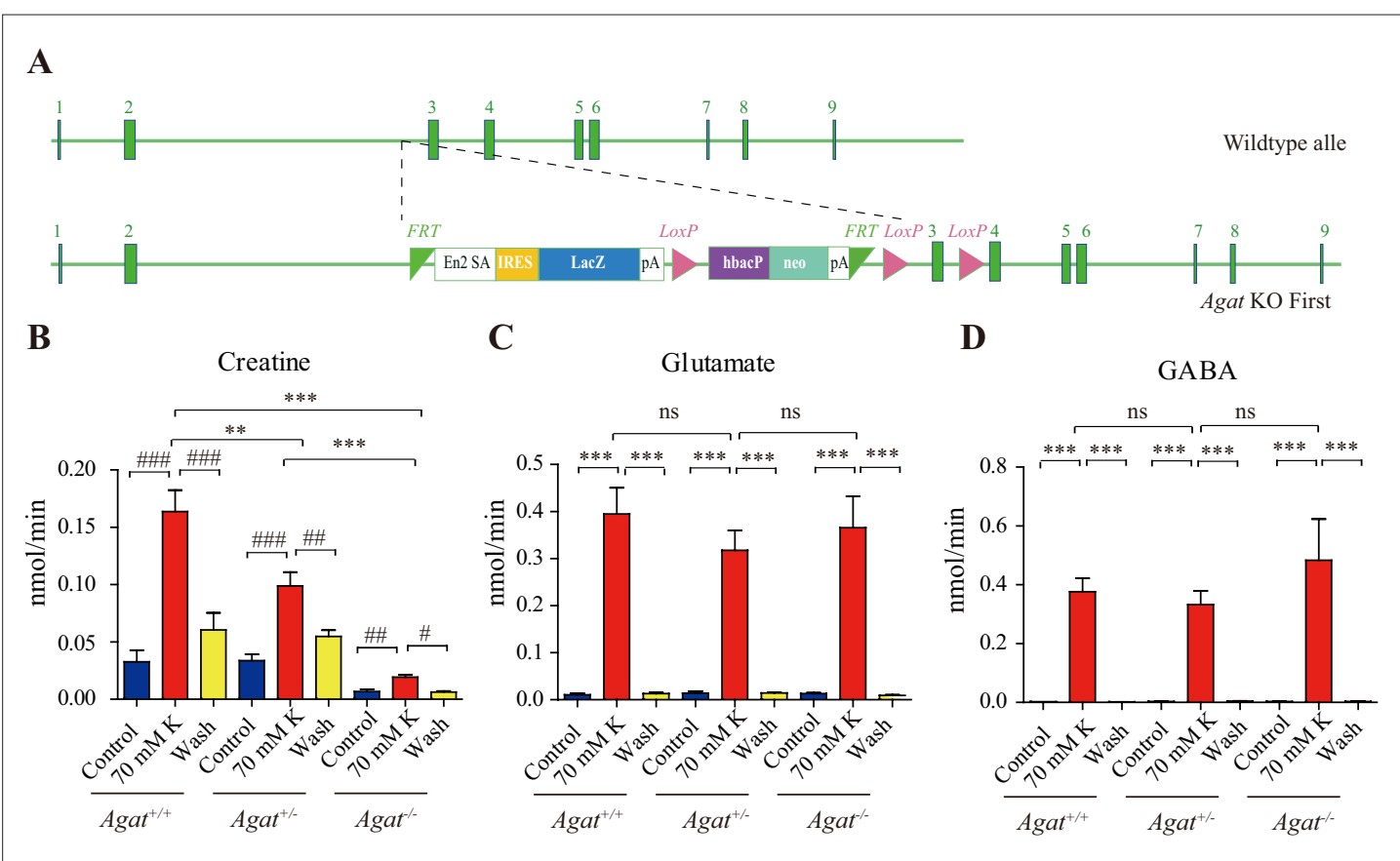

**Figure 6.** Creatine (Cr) release in WT and arginine:glycine amidinotransferase (AGAT) knockout mice. (**A**) A schematic diagram illustrating the strategy of *Agat* knockout first. With the *Agat* gene (also known as *Gatm*) shown in the upper part, and the gene targeting strategy in the lower part. The homologous arm is approximately 10 kb. A targeting cassette, containing Frt-flanked lacZ and neomycin, was inserted downstream of exon 2. At the same time, exon 3 of *Agat* was flanked by two loxP sites. $K^+$-induced release of glutamate (Glu) (**C**) and gamma-aminobutyric acid (GABA) (**D**) were not significantly different among *Agat$^{+/+}$*, *Agat$^{+/-}$*, and *Agat$^{-/-}$* mice, whereas that of Cr (**B**) was significantly lower in AGAT$^{-/-}$ mice than those in *Agat$^{+/+}$* and *Agat$^{+/-}$*.

The online version of this article includes the following source data for figure 6:

**Source data 1.** Data for *Figure 6B–D*.

slices of *Agat*^(+/-), the efflux rate in *Agat*^(-/-) mice was reduced to less than 10% that in *Agat*^(+/+) mice and 20% that in *Agat*^(+/-).

## Cr inhibition of neocortical neurons

Our own data (**Figure 4H**) and previous reports (**Mak et al., 2009**; **Lowe et al., 2015**) have shown SLC6A8 in the neocortex, with dense SLA6A8-HA immunoreactive fibers in layer 4 (**Figure 7—figure supplement 1A**; **Eliseeva and Durinian, 1975**; **Yamawaki et al., 2021**). Layer 5 neurons in the somatosensory cortex have been reported to express SLC6A8 previously (**Mak et al., 2009**; **Lowe et al., 2015**). To investigate electrophysiological effects of Cr, we performed whole-cell patch-clamp recordings from the pyramidal neurons in layer 4/5 of the somatosensory cortex (**Figure 7A and B**, **Figure 7—figure supplement 1**).

Medium-sized pyramidal neurons (**Figure 7B**) with a membrane capacitance (Cm) of 114.96 ± 3.92 pF (n = 51, **Figure 7—figure supplement 2A**) were recorded. These neurons exhibited regular firing patterns (**Scala et al., 2019**; **Stumpf et al., 2018**) in response to depolarization current injection (**Figure 7D**, **Figure 7—figure supplement 3A**) with moderate maximal evoked spiking frequencies of 10–30 spikes per 500 ms (**Figure 7E–G**), increasing of inter-spike intervals during depolarizing steps (**Figure 7D**, **Figure 7—figure supplement 3A**), high action potential amplitude (81.64 ± 1.06 mV, **Figure 7—figure supplement 2E**), and large spike half-width (1.12 ± 0.031 ms, **Figure 7—figure supplement 2F**).

Cr was bath-applied only after the evoked firing pattern reached a steady state. Of the 51 neurons, 16 were inhibited by 100 μM creatine (**Figure 7B–F**). Fewer spikes were evoked in Cr-responsive neurons in response to depolarizing current injections during Cr application (25 pA step, 500 ms) (**Figure 7D–F**). The inhibitory effect of Cr was reversible (**Figure 7D–F**), typically observed within 2–3 min following Cr application (with maximal effect from 2 to 8 min) and disappeared after 10–25 min washout. This could be repeated by a second application of Cr. The rheobase, defined as the minimal electrical current necessary to elicit an action potential, was increased during bath application of Cr (**Figure 7H**). The inhibitory effect was most obvious at near spike threshold. When a neuron was depolarized with a current of 50 pA above rheobase, the number of evoked spikes was decreased dramatically during Cr application (**Figure 7I**). Cr also mildly inhibited the input resistance (**Figure 7J**), slightly hyperpolarized resting membrane potential (**Figure 7—figure supplement 2C**), or reduced amplitude of afterhyperpolarization (AHP) followed by the first evoked action potential (**Figure 7—figure supplement 2G**). The spike threshold (**Figure 7—figure supplement 2D**), amplitude (**Figure 7—figure supplement 2E**) and half width (**Figure 7—figure supplement 2F**) were not changed by Cr.

The remaining 35 neurons were not responsive to Cr (**Figure 7C and G**, **Figure 7—figure supplement 3**). Cr did not change electrical parameters tested, including evoked firing rates (**Figure 7G**, **Figure 7—figure supplement 3A, B and D**), rheobase (**Figure 7—figure supplement 2C**), resting membrane potential (**Figure 7—figure supplement 2C**), spike threshold amplitude (**Figure 7—figure supplement 2E**), and half width (**Figure 7—figure supplement 2F**). In addition, electrical properties of responsive neurons and unresponsive neurons were not significantly different. With the limited number of neurons recorded, the ratio of responsive neurons appeared higher in layer 4 or border of layer 4/5, than the deeper layer in layer 5 (**Figure 7—figure supplement 1**).

## SLC6A8-dependent uptake of Cr into the synaptosomes

Along with enzymatic degradation, reuptake by transporters serves as an important way to remove neurotransmitters released into the synaptic cleft. As synaptosomes contain the apparatus for neurotransmission, they are often used for studying uptake of neurotransmitters (**Gulyássy et al., 2020**).

To investigate whether Cr uptake into synaptosomes required SLC6A8, we first examined whether SLC6A8 was present in synaptosomes. Using *Slc6a8*^(HA) knockin mice and an anti-HA antibody, we found that Slc6a8-HA was present and enriched in crude synaptosomal fraction (P2 fraction in **Figure 8A**, enrichment score: P2/H = 1.76 ± 0.15, n = 4) and synaptosomal fraction prepared using a discontinuous Ficoll gradient (Sy and 4-Sy fractions in **Figure 8A**, enrichment score: Sy/H = 2.02 ± 0.14, n = 4). The integrity of synaptosomes was confirmed by multiple markers of the synaptosomes (**Gulyássy et al., 2020**), including the presynaptic membrane marker SNAP25 (**Antonucci et al., 2016**) and the SV marker Syp (synaptophysin Syp) (**Jahn et al., 1985**; **Wiedenmann and Franke, 1985**; **Leube et al.,**

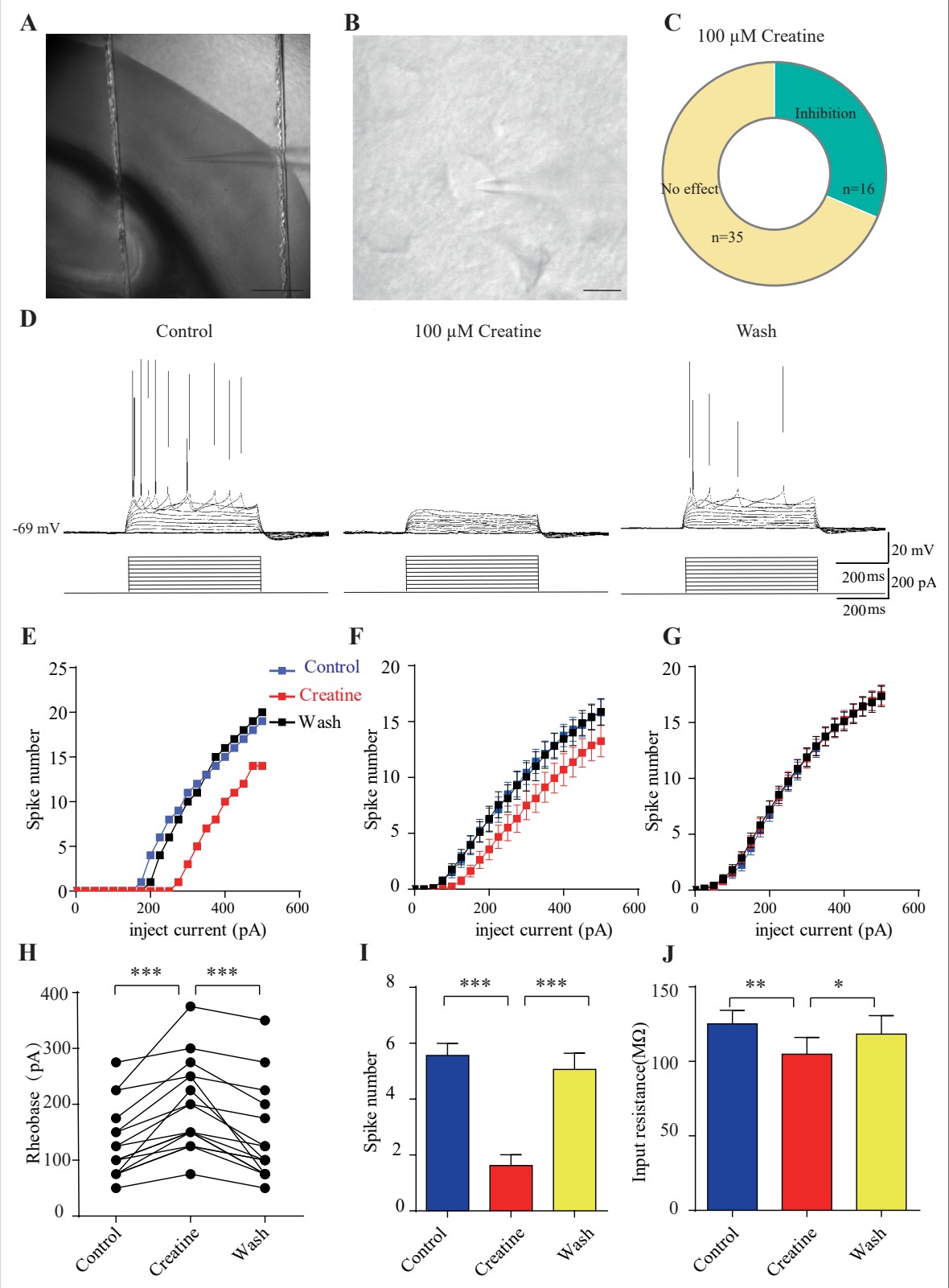

**Figure 7.** Inhibitory effects of creatine (Cr) on cortical neurons. (**A**) A photograph showing recording at layer 4 in the somatosensory cortex. Scale bar: 10 μm. (**B**) Patch-clamp recording of a pyramidal neuron. Scale bar: 10 μm. (**C**) Ratios of Cr-responsive and -unresponsive neurons in the region. (**D**) Representative raw electrophysiological traces showing inhibition of evoked firing by Cr, with the lower panel showing the stimulus protocol. (**E**) Evoked spike numbers in response to different current injections from (**D**). (**F**) Relationship between evoked spike numbers and different current

*Figure 7 continued on next page*

*Figure 7 continued*

injections to neurons that were inhibited by Cr (n = 16). (**G**) The same for Cr-unresponsive neurons (n = 35). (**H**) Rheobase for Cr-responsive neurons. (**I**) Evoked spike number when these neurons were injected with current of rheobase + 50 pA. (**J**) input resistance. *p<0.05; **p<0.01; ***p<0.001, paired *t*-test.

The online version of this article includes the following source data and figure supplement(s) for figure 7:

**Source data 1.** Data for *Figure 7E–J*.

**Figure supplement 1.** Layer distribution of recording sites in the somatosensory cortex.

**Figure supplement 2.** Electrophysiological parameters of creatine (Cr)-responsive (blue) and Cr-nonresponsive neurons (red).

**Figure supplement 2—source data 1.** Data for *Figure 7—figure supplement 2A–G*.

**Figure supplement 3.** Electrophysiological data of creatine (Cr)-unresponsive neurons.

**Figure supplement 3—source data 1.** Data for *Figure 7—figure supplement 3B–D*.

*1987*; *Südhof et al., 1987*) in presynaptic terminals, the postsynaptic density marker PSD95 (*Woods and Bryant, 1991*; *Cho et al., 1992*) and the postsynaptic membrane protein GluN1 (*Cserép et al., 2012*), the synaptic membrane protein SNAP23 (*Kunii et al., 2021*; *Ravichandran et al., 1996*; *Suh et al., 2010*), the plasma membrane marker Na$^+$-K$^+$-ATPase (*Cameron et al., 1994*; *Mersel et al., 1987*; *Sun et al., 1988*), and the mitochondria marker VDAC (*Gulyássy et al., 2020*). These were all enriched in our synaptosomal preparations. The cytosol marker GAPDH was also present in synaptosomes, whereas the oligodendrocyte marker MBP was nearly absent, suggesting that myelin pollution was largely avoided (*Gulyássy et al., 2020*).

We have also used EM to confirm the quality of our synaptosome preparations. As reported previously (*Gulyássy et al., 2020*; *Schrimpf et al., 2005*), synaptosomes were composed of membrane bounded structures (Sy in *Figure 8B*) filled with synaptic vesicles (SV in *Figure 8B*), sometimes with a segment of postsynaptic membrane along with the postsynaptic density (PSD in *Figure 8B*) and mitochondria (Mt in *Figure 8B*). The sizes of synaptosomes from WT mice and *Slc6a8* knockout mice were similar, with areas of 0.245 ± 0.01 μm$^2$ (n = 302 particles) and 0.247 ± 0.01 μm$^2$ (n = 317 particles), respectively (*Figure 8B*).

We then examined whether SLC6A8 participated in Cr uptake into the synaptosomes. A mixture of 18 μM [$^{14}$C]-Cr (with a total radioactivity of 0.4 μCi) and 5 μM Cr was used, and uptake at 0°C measured at 10 min was the baseline (*Fykse and Fonnum, 1988*). Cr uptake into synaptosomes from WT mice was stimulated approximately sevenfold at 37°C (Uptake, *Figure 8C*) compared to 0°C (Ctrl, *Figure 8C*). Cr uptake into synaptosomes from *Slc6a8* knockout mice was less than three times compared to its control, and was decreased to approximately 1/3 of that of WT mice (*Figure 8C*). Thus, SLC6A8 is necessary for uptake of Cr into the synaptosomes.

## Cr uptake into SVs

Classical neurotransmitters were taken up in SVs in an ATP-dependent manner (*Burger et al., 1991*; *Cidon and Sihra, 1989*; *Schenck et al., 2009*; *Fykse and Fonnum, 1988*; *Bellocchio et al., 2000*). We examined whether Cr could be transported into SVs.

We used 10 μg anti-Syp antibody to purify SVs from mouse brains. Purified SVs were preincubated for 30 min to allow sufficient leakage of endogenous Cr, before being mixed with 1 mM [$^{13}$C]-Cr in the presence or absence of 4 mM ATP and placed at 25°C for 10 min to allow adequate uptake. The SV content of [$^{13}$C]-Cr was then examined by CE-MS and high-performance liquid chromatography-mass spectrometry (HPLC-MS). Significantly more [$^{13}$C]-Cr were taken up by SVs in the presence of ATP, with about 10.3 pmol [$^{13}$C]-Cr transported into SVs (1.03 pmol/μg α-Syp or transportation rate of 0.103 pmol/min, n = 11, *Figure 8D*).

In summary, Cr could be transported into SVs in an ATP-dependent manner. At this point, we do not know what is the transporter(s) on the SVs for Cr uptake. SLC6A8 is only found in plasma membrane, not on SVs, and is not a candidate for Cr uptake into SVs.

## Discussion

While no neurotransmitter has been proven in a single paper, supportive evidence suggesting Cr as a possible new neurotransmitter has been presented here to the extent of any single previous papers.

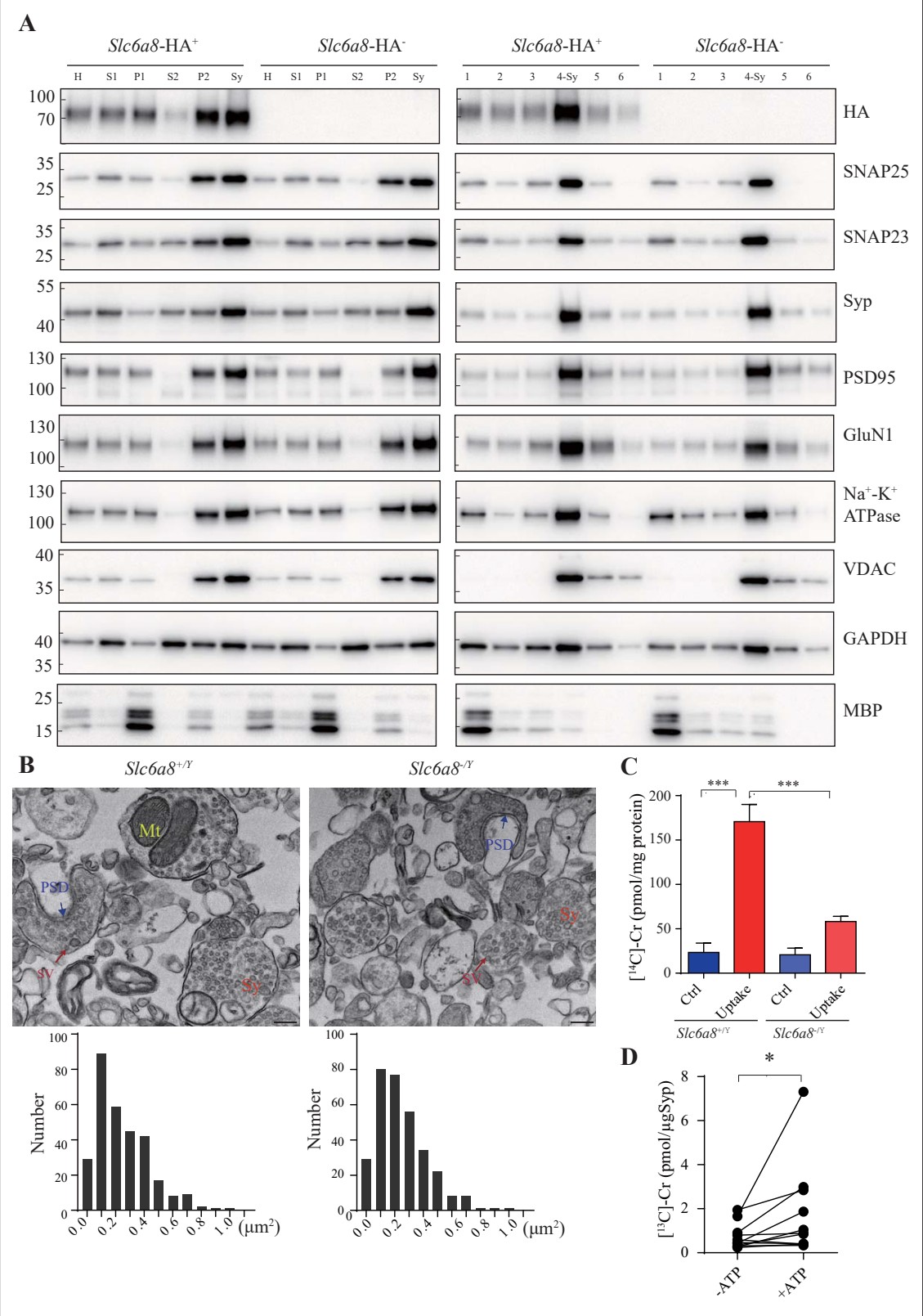

**Figure 8.** Creatine (Cr) uptake into synaptosomes and synaptic vesicles (SVs). (**A**) Markers for subcellular organelles detected in synaptosomes prepared from WT mice or mice with *Slc6a8* gene fused in frame to the HA epitope. SLC6A8-HA was enriched in synaptosomes (Sy or 4-Sy) or crude synaptosomes (P2) (see 'Materials and methods'), as were other markers for the subcellular organelles in synaptosomes but not the marker for myelin (MBP). (**B**) Representative electron micrographs and histograms of size distribution in synaptosomes isolated from WT (*Slc6a8*[+/Y]) and *Slc6a8* KO

*Figure 8 continued on next page*

*Figure 8 continued*

($Slc6a8^{-/Y}$) mice by Ficoll density-gradient centrifugation. Sy, synaptosome; Mt, mitochondria; PSD, postsynaptic density. Bar, 20 nm. (**C**) Cr uptake into synaptosomes (n = 5 per group). The two left columns were results from WT mice and the two right columns from Slc6a8 knockout mice. The control baseline was [$^{14}$C]-Cr uptake at 0°C at 10 min. Cr uptake into synaptosomes at 37°C measured at 10 min was observed in WT synaptosomes. Uptake into *Slc6a8* knockout synaptosomes was significantly reduced compared to the WT synaptosomes. \*\*\*$p<0.001$, one-way ANOVA with Tukey's correction. (**D**) uptake of [$^{13}$C]-Cr into immunoisolated SVs in the presence or absence of ATP (n = 11 samples per group). \*$p<0.05$, paired *t*-test.

The online version of this article includes the following source data for figure 8:

**Source data 1.** Original western blot files for *Figure 8A*.

**Source data 2.** Labelled western blot files for *Figure 8A*.

**Source data 3.** Labelled western blot files for *Figure 8A*.

At various times and by different researchers, taurine (*Curtis and Watkins, 1960*; *Saransaari and Oja, 2008*), proline (*Felix and Künzle, 1974*), D-aspartic acid (*D'Aniello et al., 2011*), hydrogen sulfide (*Abe and Kimura, 1996*), agmatine (*Reis and Regunathan, 2000*), DOPA (*Misu et al., 2002*), estradiol (*Balthazart and Ball, 2006*), β-alanine (*Tiedje et al., 2010*), and protons (*Du et al., 2014*) have been suspected as neurotransmitters, but they do not meet all the criteria. Some of the suspected molecules can be released upon stimulation or removed by transporters. Often, they have not been reproducibly found in SVs (*Chantranupong et al., 2020*).

Our discovery of Cr in SVs significantly raised the priority of testing the candidacy of Cr, and our further investigations have led to more evidence suggesting Cr as a neurotransmitter: (1) Cr is stored in SVs; (2) $Ca^{2+}$-dependent release of Cr upon stimulation has been observed; (3) both Cr storage in SVs and Cr release are reduced when either the gene for *Slc6a8* or the gene for *Agat* was deficient; and (4) Cr inhibits activities of pyramidal neurons in the neocortex; (5) Cr uptake into synaptosomes requires SLC6A8; and (6) Cr uptake into SVs was ATP-dependent.

Of the above results, 1, 3, 4, and 6 are reported for the first time in this article. Furthermore, we have demonstrated that detection of Cr in SVs was lower than those for Glu and GABA, but higher than those for ACh and 5-HT, placing Cr at a level in the middle of known central transmitters (*Figures 1–3*). The storage of Cr in SVs is dependent on preserved $H^+$ gradient (*Figure 1—figure supplement 2*) and Cr can be transported into SVs (*Figure 8D*).

There was a single previous report of $Ca^{2+}$-dependent release of [$^3$H]Cr and endogenous Cr in response to electrical stimulation (*Almeida et al., 2006*). We now provide evidence that Cr was released in response to extracellular $K^+$ stimulation (within 1–2 min) (*Figures 5 and 6*). Furthermore, Cr release was reduced when either the *Slc6a8* or *Agat* gene was removed (*Figures 5 and 6*). Although the $Ca^{2+}$-dependent component of $K^+$-evoked Cr release was smaller compared to those of Glu and GABA, it nevertheless existed and was totally abolished by *Slc6a8* knockout (*Figure 5*). The reported electrically evoked Cr release showed more $Ca^{2+}$ dependence (*Almeida et al., 2006*). Taken together, our data and previous report (*Almeida et al., 2006*) supported a role of Cr as a neurotransmitter. Our observation of extremely low efflux rates of 5-HT or ACh may have arisen from very limited numbers of cholinergic (*Li et al., 2018*) or serotoninergic neurons (*Ishimura et al., 1988*) in the sliced sections and rapid enzymatic degradation of these neurotransmitters.

Cr uptake from the extracellular space into the cells was reported twice previously, once with brain slices showing sodium-dependent uptake of [$^3$H]Cr (*Almeida et al., 2006*) and once with synaptosomes (*Peral et al., 2010*). Our new results have not only replicated the synaptosome Cr uptake experiment but also shown the requirement of SLC6A8, a membrane transporter expressed in synaptosomes (*Figure 8A*), for Cr uptake into synaptosomes. Transportation of Cr into synaptosomes by Slc6A8 may function for both the clearance of Cr from the synaptic cleft and recycling of Cr into SVs residing in neurons (*Figures 6D and 8B*).

In summary, in addition to confirming and extending previous results which have stood alone for more than a decade without replication or follow-up, we have obtained entirely new results suggesting the candidacy of Cr as a neurotransmitter. We discuss below the criteria for a neurotransmitter, Cr as a neurotransmitter, and the implications of Cr as a neurotransmitter.

## Criteria of a neurotransmitter

The criteria for establishing a non-peptide small molecule as a neurotransmitter have varied from time to time and from author to author.

Some textbooks simply state that a neurotransmitter is stored presynaptically, released upon stimulation, and active on postsynaptic neurons. The details of these three criteria can vary. For example, one textbook stipulates that "the substance must be present within the presynaptic neuron; the substance must be released in response to presynaptic depolarization, and the release must be $Ca^{2+}$ dependent; specific receptors for the substance be present on the postsynaptic cell" (*Purves et al., 2001*; *Purves et al., 2016*). Another states that "the molecule must be synthesized and stored in the presynaptic neuron; the molecule must be released by the presynaptic axon terminal upon stimulation; the molecule, when experimentally applied, must produce a response in the postsynaptic cell that mimics the response produced by the release of neurotransmitter from the presynaptic neuron" (*Bear et al., 2016*).

The neuroscience textbook most widely used internationally for the last four decades lists four criteria for a neurotransmitter (*Kandel et al., 2013*; *Kandel et al., 2021*): it is synthesized in the presynaptic neuron; it is present within vesicles and is released in amounts sufficient to exert a defined action on the postsynaptic neuron or effector organ; when administered exogenously in reasonable concentrations, it mimics the action of the endogenous transmitter; and a specific mechanism usually exists for removing the substance from the synaptic cleft. These are similar, but not identical, to the classic textbook on neurotransmitters: a neurotransmitter "should be synthesized and released presynaptically; it must mimic the action of the endogenous compound that is release on nerve stimulation; and where possible, a pharmacological identity is required where drugs that either potentiate or block postsynaptic responses to the endogenously released agent also act identically to the suspected neurotransmitter that is administered" (*Cooper et al., 2002*). The pharmacological criterion is listed in another textbook (*Squire et al., 2012*).

Some authors note difficulties in establishing a CNS neurotransmitter. For example, a specialized neurotransmitter book states that "the candidate neurotransmitter should be present in the presynaptic terminal, be released when the presynaptic terminal is active, and when applied experimentally, induce faithful responses in the postsynaptic neuron. In practice, since central nervous system neurons continuously integrate diverse excitations and inhibitions, the last criterion is relaxed to demonstrating merely changes in such activity" (*Robinson, 2001*).

Solomon Snyder, a leading scientist of classic neurotransmitters, neuropeptides and their receptors, wrote that "designating a molecule as a transmitter depends on the criteria employed, the most common of which are that the substance is synthesized in neurons, released by their terminals, mimics the effects of physiologic neurotransmission and possess a mechanism for inactivation. However, with each new candidate the rules have been modified and broadened" (*Barañano et al., 2001*).

## Evidence supporting Cr as a neurotransmitter

Sixteen small molecules have been listed as neurotransmitters in the classic textbook (*Kandel et al., 2013*; *Kandel et al., 2021*). Among them, adenosine, arachidonic acid, nitric oxide, and carbon monoxide do not meet all four criteria at present. Cr appears to be better than these in meeting the criteria for a central neurotransmitter.

The results obtained by us in this article have satisfied the criteria of *Robinson, 2001* for Cr to be a CNS neurotransmitter.

The four criteria of Snyder and colleagues (*Barañano et al., 2001*) have been mostly met but the physiological neurotransmission would require more research because a specific synapse(s) would have to be defined and studied for putative creatinergic neurotransmission. This can take much longer in the CNS than the PNS. Some commonly accepted neurotransmitters have never satisfied this criterion in a strict sense. The mechanism of Cr removal criterion is met not only by the Cr uptake in brain slices (*Almeida et al., 2006*) and in synaptosomes (*Peral et al., 2010*), but also by our demonstration that SLC6A8 is required for synaptosome uptake of Cr.

The four criteria of *Kandel et al., 2013* and *Kandel et al., 2021* are mostly satisfied with some details requiring further research. The synthesis requirement is usually not strict because there are transmitters synthesized in some cells and transported into others where they function as transmitters. Our discovery of Cr in SVs can replace the synthesis requirement because the presence in neuronal SVs provide sufficient evidence that Cr is located in the right location to function as a neurotransmitter. The level of Cr in SVs is higher than those of ACh and 5-HT (*Figures 1 and 2*). The amount of released Cr is in the same order of magnitude as those of Glu and GABA (*Figures 5 and 6*). The criterion of a

specific mechanism of removal was met by Cr uptake experiments in slices (*Almeida et al., 2006*) and in synaptosomes (*Peral et al., 2010*), and further strengthened by our finding of SLC6A8 involvement in synaptosome uptake of Cr (*Figure 8*).

Here, we report that Cr, at a concentration comparable to classical neurotransmitters, inhibits pyramidal neurons in specific regions of the mouse brain, with approximate 1/3 of pyramidal neurons responding to 100 µM Cr (*Figure 7*). In previous reports, 100 µM to 2 mM of GABA (*Brown and Scholfield, 1979*; *Pringle et al., 1996*; *Osmanović and Shefner, 1990*), 50 µM to 2 mM of Glu (*Akaike et al., 1987*; *Cole et al., 1989*), 1–100 µM of DA (*Akaike et al., 1987*), and 0.1–100 µM of 5-HT (*Beck and Goldfarb, 1985*; *Reynolds et al., 1988*; *Mendiguren et al., 2022*; *Xie et al., 2012*; *Gorelova and Reiner, 1996*; *Okuhara and Beck, 1994*) were bath-applied to investigate the physiological functions of neurotransmitters. Our results revealed that, when bath-applied, Cr could inhibit cortical neurons at 100 µM within several minutes, with a time course similar to that of 5-HT (*Sizer et al., 1992*; *Xie et al., 2012*; *Okuhara and Beck, 1994*) and DA (*Akaike et al., 1987*), but significantly slower than that of Glu (*Cole et al., 1989*), GABA (*Roberts and Frankel, 1950*; *Pringle et al., 1996*), and 5-HT (*Mendiguren et al., 2022*).

In a recent report, knockout of the Slc6a8 gene increased excitation of cortical neurons (*Ghirardini et al., 2023*). Electrophysiological characterization of pyramidal neurons in the prefrontal cortex (PFC) found increased evoked firing frequency. Because we have shown that Cr inhibit a fraction of pyramidal neurons in the neocortex (*Figure 8*), this article provides in vivo evidence consistent with the possibility of Cr as an inhibitory neurotransmitter.

## Differences between Cr and classic neurotransmitters

At this point, we do not have a molecularly defined receptor for Cr, only inferring its presence from the electrophysiological responses to Cr. We speculate that Cr may act on G-protein-coupled receptors (GPCRs), rather than the fast-acting ligand-gated ion channels, such as AMPA or NMDA receptors for Glu and GABA$_A$ receptor for GABA. There have been previous reports of Cr effects on neurons, including Cr as a partial agonist for GABA$_A$ receptors (*De Deyn and Macdonald, 1990*; *De Deyn et al., 2001*; *Neu et al., 2002*; *Koga et al., 2005*). These effects require very high concentrations of Cr (in the 10 mM range). There was also a report of the opposite effect: that Cr (at a concentration above 500 µM) increased neuronal excitability through NMDA receptors after incubation for 60 min, with a time course significantly slower than those of classic neurotransmitters (*Royes et al., 2008*).

Ca$^{2+}$-independent component of Cr release induced by extracellular K$^+$ was more prominent than those of Glu or GABA. One possibility was that Ca$^{2+}$-independent Cr release came from glia because high GAMT levels were reported in astrocytes (*Schmidt, 2004*) and oligodendrites (*Schmidt, 2004*; *Rosko et al., 2023*). As reported, other neuromodulators such as taurine can be released from astrocytes (*Philibert et al., 1989*) or slices (*Saransaari and Oja, 2006*) in a Ca$^{2+}$-independent manner. In addition, in the absence of potassium stimulation, Ca$^{2+}$ depletion increased release of taurine in cultured astrocytes (*Takuma et al., 1996*) or in striatum in vivo (*Molchanova et al., 2005*). Similarly, in *Slc6a8* KO slices, Ca$^{2+}$ depletion (*Figure 5G*) also increased Cr baseline compared to that in normal ACSF (*Figure 5D*).

With much longer history of research, ACh and 5-HT now have more evidence in other aspects than Cr as a central transmitter, especially because there are many agonists and antagonists for ACh and 5-HT to prove an additional criterion that is required in some (*Cooper et al., 2002*; *Squire et al., 2012*), but not the majority of, textbooks for a neurotransmitter. The pharmacology criterion will take some time and effort because so far no effort has been made to find agonists or antagonists for Cr.

## Implications of SLC6A8 and Cr

It is notable that SLC6A8 belongs to the NTT family, with multiple members already shown to transport neurotransmitters (*Pacholczyk et al., 1991*; *Blakely et al., 1991*; *Guastella et al., 1992*; *Clark et al., 1992*; *Borden et al., 1992*; *Lopez-Corcuera et al., 1992*; *Giros et al., 1991*; *Kilty et al., 1991*; *Shimada et al., 1991*; *Hoffman et al., 1991*; *Smith et al., 1992*; *Liu et al., 1992*).

The uptake experiments by others and us indicate that SLC6A8 transports Cr into neurons within the brain. AGAT is also expressed in the brain, but in cells not expressing SLC6A8 (*Braissant and Henry, 2008*; *Braissant et al., 2010*). Cr and its precursor were thought to be transported between different cells in the nervous system. When SLC6A8 was completely missing, such as in homozygous

*SLC6A8*-deficient patients, Cr treatment was not effective. But if SLC6A8 was partially active, Cr was effective (**Dunbar et al., 2014**). Intractable epilepsy in a female with heterozygous *SLC6A8* mutation was completely treated by Cr (**Mercimek-Mahmutoglu et al., 2010**). Our data of inhibitory effect of creatine on cortical neurons might provide a new mechanism to its anti-epileptic activity (**Gerbatin et al., 2019**).

The absence of SLC6A8 expression in astrocytes whose endfeet lining microcapillary endothelial cells (MCEC) form the blood–brain barrier (BBB) indicates that Cr in the brain does not rely on import from the periphery and is instead mainly synthesized in the brain (**Braissant et al., 2011**; **Braissant et al., 2001**; **Braissant, 2012**). SLC6A8 functions within the brain to transport Cr and its precursors not as a major contributor of Cr transport across the BBB. It is thought to mediate Cr uptake into the presynaptic terminal based on studies of synaptosomes (**Figure 8**; **Peral et al., 2010**).

Cr is known to have effects other than an energy source, and Cr supplement has been thought to be beneficial for children, pregnant and lactating women, and old people (**Wallimann et al., 2011**; **Brosnan and Brosnan, 2016**). Cr has been reported to improve human mental performance (**Watanabe et al., 2002**; **Rae et al., 2003**; **McMorris et al., 2006**; **McMorris et al., 2007**; **Rae and Bröer, 2015**; **Wallimann and Harris, 2016**). Cr has been used as potential treatment in animal models of neurodegenerative diseases (**Andreassen et al., 2001**; **Andres et al., 2005**).

Our work will stimulate further research to distinguish which of the previously suspected effects of Cr is not attributed to its role as an energy storage, but can be attributed to its role as a neurotransmitter.

## Search for new neurotransmitters

Our work may stimulate the search for more neurotransmitters. Our discovery indicates that the hunt for neurotransmitters stopped decades ago because of technical difficulties not due to the absence of more neurotransmitters. The fact that most of the known small-molecule neurotransmitters have been found because of their peripheral effects also argues that what is missing is the concerted efforts to uncover central neurotransmitters with no peripheral effects. New neurotransmitters may be discovered from candidates which have been long suspected and from previously unsuspected molecules or even previously unknown molecules.

Innovative approaches should be taken to uncover molecules with no previous suspicions or hints. Highly purified SVs, SVs from different regions of the brain, and SVs with specific SLCs offer some of the starting points for future research.

# Materials and methods
## Generation of knockout and knockin mice

*Slc6a8* knockout and knockin mice were generated using CRISPR-Cas9-mediated genome engineering techniques by Beijing Biocytogen (Beijing, China). *Agat* 'knockout-first' (**Skarnes et al., 2011**) mice were purchased from CAM-SU GRC (Suzhou, China). All mutations were validated by Southern blot analysis, tail junction PCR, and DNA sequencing. Transgenic mice will be provided upon request.

## RT-PCR and qPCR

Total RNA of whole brains from mice of different genotypes was extracted using the Buffer RZ (Tiangen, no. RK14, Beijing, China) and reverse transcribed into complementary DNA (cDNA) using the RevertAid First-Strand cDNA synthesis kit (Thermo Scientific, K1622). qPCR was performed using the Taq Pro Universal SYBR qPCR Master Mix (Vazyme, Q712-02) on Bio-Rad CFX-96 Touch Real-time PCR System (Bio-Rad, USA). Glyceraldehyde-3-phosphate dehydrogenase (*Gapdh*) was used as an internal control. ΔCt (difference in cycle threshold) was calculated for each sample (ΔCt = Ct $_{Target\ gene}$ − Ct $_{GAPDH}$) for further evaluation of relative mRNA expression levels in different genotypes. The sequence specificities of the primers were examined. Three pairs of primers targeting different genes were used: *Slc6a8* forward, 5'-GTCTGGTGACGAGAAGAAGGG-3', *Slc6a8* reverse, 5'-CCACGCAC GACATGATGAAGT-3'; *Agat* forward, 5'-cacagtggaggtgaaggccaatacatat-3', *Agat* reverse, 5'-ccgcctca cggtcactcct-3'; *Gapdh* forward, 5'- AGGTCGGTGTGAACGGATTTG-3', *Gapdh* reverse, 5'-TGTAGACC ATGTAGTTGAGGTCA-3'.

Primers for reverse PCR were designed to obtain complete coding sequences based on information obtained from the National Center for Biotechnology Information (NCBI): *Slc6a8* forward,

5′-atggcgaaaaagagcgctgaaaacg-3′; *Slc6a8* reverse, 5′-ttacatgacactctccaccacgacgacc-3′; *Agat* forward, 5′- atgctacgggtgcggtgtct-3′; *Agat* reverse, 5′-tcagtcaaagtaggactgaagggtgcct-3′. PCR products were electrophoresed on 1% agarose gels, stained with GelRed, visualized under UV illumination, and photographed.

## Immunoblot analysis

Samples were loaded onto 10% polyacrylamide gels with the PAGE system (#1610183, Bio-Rad Laboratories, USA) and run in the SDS running buffer (25 mM Tris, 192 mM glycine, 0.1% SDS, pH 8.8) for 25 min at 80 V followed by 25–45 min at 200 V. Afterward, proteins were transferred to immobilon NC transfer membranes (HATF00010, Millipore) at 400 mA for 2 hr in transfer buffer (25 mM Tris, 192 mM glycine, 20% methanol). Membranes were blocked in 5% fat-free milk powder in TBST (25 mM Tris, 150 mM NaCl, 0.2% Tween-20 [P1397, Sigma, St. Louis, MO], pH 7.4 adjusted with HCl) and incubated overnight with the indicated primary antibodies dissolved in TBST containing 2% BSA.

Primary antibodies are listed below: rabbit anti-synaptophysin (dilution 1:5000, cat. no. 101002, synaptic systems [SySy], Goettingen, Germany), rabbit anti-synaptotagmin1/2 (dilution 1:2000, cat. no. 105002, SySy), rabbit anti-proton ATPase (dilution 1:1000, cat. no. 109002, SySy), rabbit anti-synaptobrevin 2 (dilution 1:5000, cat. no. 104202, SySy), rabbit anti-SV2A (dilution 1:2000, cat. no. 109003, SySy), rabbit anti-VGlut1 (dilution 1:4000, 135302, SySy), rabbit anti-VGlut2 (dilution 1:2000, 135402, SySy), rabbit anti-VGAT (dilution 1:4000, 131002, SySy), rabbit anti-SNAP23 (dilution 1:2000, cat. no. 111202, SySy), mouse anti-PSD95 (dilution 1:5000, cat. no. 75028, NeuroMab, Davis, CA), mouse anti-GluN1(dilution 1:5000, cat. no. 114011, SySy), rabbit anti-GM130 (dilution 1:1000, cat. no. ab52649, Abcam), rabbit anti-Golgin-97 (dilution 1:2000, cat. no. 13192, Cell Signaling Technology, MA), rabbit anti-EEA1 (dilution 1:2000, cat. no. 3288, Cell Signaling Technology), rabbit anti-LC3B (dilution 1:1000, cat. no. 2775S, Cell Signaling Technology), goat anti-CathepsinB (dilution 1:2000, AF965, R&D Systems, Minneapolis, MN), rabbit anti-GAPDH (dilution 1:1000, cat. no. 2118S, Cell Signaling Technology), rabbit anti-GluT4 (dilution 1:1000, ab33780, Abcam, Cambridge, UK), rabbit anti-CACNA1A (dilution 1:300, 152103, SySy), rabbit anti-VDAC (dilution 1:1000, cat. no. 4661S, Cell Signaling Technology), rabbit anti-MBP (1:1000, cat. no. 295003, SySy), mouse anti-Creatine Kinase B (dilution 1:5000, cat. no. MAB9076, R&D Systems), rabbit anti-HA (dilution 1:2000, CST3724, Cell Signaling Technology), and rabbit anti SNAP25 (dilution 1:2000, ab109105, Abcam, UK) antibodies.

Membranes were washed in three washing steps in TBST (each for 5 min) and incubated with peroxidase-conjugated secondary antibodies for 2–3 hr at 4°C. The second antibodies used were anti-rabbit (dilution 1:5000, A6154, Sigma), anti-mouse (dilution 1:5000, 715-035-150, Jackson ImmunoResearch, West Grove, Philadelphia, USA) or rabbit anti-goat IgG secondary antibodies (dilution 1:1000, cat. no. ab6741, Abcam). After repeated washing, signals were visualized using a ChemiDoc XRS $^+$ System (Bio-Rad Laboratories).

## Isolation of synaptic vesicles

Our purification procedures for SVs were based on previously established immunoisolation methods (*Burger et al., 1989*; *Martineau et al., 2013*). Protein G magnetic beads (cat. no. 88848, Thermo Fisher Scientific, Waltham, MA) were washed three times with IP buffer (100 mM potassium tartrate, 4 mM HEPES-KOH, 2 mM MgCl$_2$, pH 7.4) supplemented with a complete protease inhibitor cocktail (Roche, Basel, Switzerland). Then, 5 µg monoclonal anti-Syp antibody directed against a cytoplasmic epitope (cat. no. 101011, SySy) or control mouse IgG (10400C, Thermo Fisher Scientific) was used to incubate with 20–30 µl beads for 30 min at RT in 2% BSA dissolved in IP buffer. Under this condition, 4–4.5 µg of antibody was coupled, as determined by western blot and Coomassie Blue staining. Immunoisolation of SVs was carried out at 0–2°C to prevent vesicular content leakage (with RT as a control). Briefly, the whole mouse brain was homogenized in 3 ml of IP buffer with a glass/Teflon homogenizer (20 strokes at 2000 rpm, WHEATON, USA, and WIGGENS WB2000-M, Germany) immediately after decapitation. Homogenates were centrifuged for 25 min at 35,000 × *g*, and the supernatant was adjusted to approximately 3 mg/ml protein (NanoDrop 2000C, Thermo Fisher Scientific). To capture the SVs for content detection, about 200 µl of supernatants (per 5 µg anti-Syp/IgG) was incubated with pre-coupled beads for 2.25 hr under slow rotation at 2°C. Beads were washed six times for further western blot analysis and vesicular content detection. For pharmacological blockade

of H$^+$-gradient across SV membrane, the mix of supernatants and pre-coupled beads was diluted into 1.2 ml before the addition of inhibitors.

## Determination of vesicular contents

To extract SV contents, immunoisolates were treated with 50 µl ultra-pure water. Then, 100 µl methanol together with 100 µl acetonitrile was added to precipitate proteins in samples. After centrifugation for 20 min at 16,8000 × $g$, supernatants were collected and centrifuged for 20 min at 2000 × $g$ to remove beads and proteins. Samples were pre-frozen with liquid nitrogen and vacuum dried at –45°C overnight. Dried samples were kept frozen and resuspended with 50 µl of 0.2 µM $^{13}$C-creatine (internal control) immediately before detection.

CE-MS was used to verify and quantify small molecules. CE/MS detection was applied with the coupling of PA800 plus CE system (Beckman Coulter, Brea, CA) and mass spectrometry (TRIPLE QUAD 5500, AB SCIEX or Q Exactive HF-X, Thermo Scientific). Before SV content detection, we optimized MS detection of classical neurotransmitters, Cr, and amino acids in positive ion mode. Firstly, the fragment ions (Q3) for a given molecule (precursor ions, Q1) were determined by either systematic scanning of standard sample solution (0.1 µM in 10% acetate acid) or referring to database (https://www.mzcloud.org). Secondly, optimal values of collision energy (CE), collision cell exit potential (CXP), and declustering potentials (DP) were determined for each pair of Q1/Q3. Thirdly, optimal combination of parameters (Q1/Q3, CE, CXP, DP) was chosen for each molecule. In addition, parameters were adjusted every 2–3 mo for best signal-to-noise ratios.

CE/MS separations were carried out by capillaries (OptiMS silica surface cartridge, Beckman Coulter). The CE background electrolyte was 10% acetate acid. Each new separation capillary was activated with rinsing under 100 psi sequentially with methanol for 10 min (forward), methanol for 3 min (reverse), H$_2$O for 10 min (forward), H$_2$O for 3 min (reverse), 0.1 M NaOH for 10 min (forward), water for 5 min (reverse), 0.1 M HCl for 10 min (forward), followed by water for 10 min and then 10% acetate acid for 10 min (forward) and 3 min (reverse), prior to the first use. Between analyses, the capillary was rinsed with 10% acetate acid under a 100 psi pressure for 5 min (forward) flowed by 75 psi for 4 min. The sample (50 µl) was injected with 2.5–4 psi for 30 s. Separation voltage of 25 kV was applied for 25 min. To maintain stably spray during CE separation, ion spray voltage was applied at 1.7–1.9 kV. MS data were collected 5 min after CE separation. Finally, the capillary was washed with 10% acetate acid for 10 min, followed by methanol for 20 min and then 10% acetate acid for 20 min.

Standard solutions of 0.2 µM $^{13}$C-Cr (internal control) and analytes were used to plot standard curves. Linear standard curves ($R^2 > 0.98$, for most cases, $R^2 > 0.99$), calculated from peak area ratios corresponding to analytes and internal standards, were obtained for all molecules tested. The concentration ranges used for standards of Glu, GABA, ACh, 5-HT, Cr, and alanine were 0.03–10 µM, 0.003–1 µM, 0.0003–0.1 µM, 0.003–1 µM, 0.03–1 µM, and 0.03–1 µM, respectively. Standard curves were made at least twice for a given capillary. Analytes of SV contents were calculated using the standard curves and normalized to the amount of anti-Syp antibody conjugated to the beads.

## Electron microscopy

All EM grids were glow discharged for 30 s using a plasma cleaner (Harrick PDC-32G-2, plasma cleaners, Ithaca, NY). To free SVs from beads, 25 µl 0.1 M glycine-HCl (PH = 2) was incubated for 1 min and quickly neutralized with 25 µl 0.1 M Tris (pH = 10). Beads were quickly removed and 2–4 µl aliquots of SVs were applied to the carbon-coated copper grids (Zhong Jing Ke Yi, Beijing, China). After 1 min, the grid was dried with a filter paper (Whatman No. 1), and placed in the water, and then immediately stained using 2% uranyl acetate for 30 s. At last uranyl acetate was removed and the grid was air dried. The grids were examined on a JEM-F200 electron microscope (JEOL, Tokyo, Japan) operated at 200 kV. Images were recorded using a 4k × 4k COMS One view camera (Gatan, Abingdon, UK). Fixation of synaptosomal pellets was performed by immersion with pre-warmed 2.5% glutaraldehyde in 0.1 M phosphate buffer (pH 7.4) at RT for 2 hr. After washing four times with 0.1 M phosphate buffer (pH 7.4) every 15 min, samples were post-fixed with 1% osmium tetroxide (w/v) at 4°C for 1 hr and then washed three times. Following en bloc staining with 2% uranyl acetate (w/v) at 4°C overnight, samples were dehydrated and embedded in fresh resin, polymerized at 65°C for 24 hr. Ultrathin (70 nm) sections were obtained using Leica UC7 ultramicrotome (Leica Microsystems,

Wetzlar, Germany) and recorded on 80 kV in a JEOL Jem-1400 transmission electron-microscope (JEOL) using a CMOS camera (XAROSA, EMSIS, Munster, Germany).

## Immunohistochemistry

Adult mice were anesthetized by i.p. injection with 2% 2,2,2-tribromoethanol (T48402, Sigma) in saline at a dose of 400 mg/kg and perfused trancardially with 0.9% saline, followed by 4% PFA in PBS (137 mM NaCl, 2.7 mM KCl, 10 mM $Na_2HPO_4$, 1.8 mM $KH_2PO_4$, pH = 7.4).

Brains were cryoprotected with 30% sucrose in 30% sucrose 0.1 M PB (81 mM $Na_2HPO_4$, 19 mM $NaH_2PO_4$) and sectioned in the coronal plane (40 μm thick) using a Cryostat (Leica 3050S). For anti-HA immunostaining, we used a rabbit monoclonal anti-HA antibody (1:500 in 0.3% Triton in PBS; 48 hr incubation at 4°C; #3724, Cell Signaling Technology), followed by a goat anti-rabbit Alexa Fluor 546 secondary antibody (1:1000; overnight at 4°C; # A-11035, Invitrogen, Waltham, MA). Sections were mounted in a medium containing 50% glycerol, cover-slipped, and sealed with nail polish. Images were acquired using virtual slide microscope (Olympus VS120-S6-W, Tokyo, Japan) and a laser-scanning confocal microscope (Zeiss 710, Cambridge, UK) and brain structures inferred with an established mouse brain atlas (*Paxinos and Franklin., 2013*).

## Preparations of brain slices

Male C57 mice (of 30–38 days old) were anesthetized with pentobarbital (250 mg/kg) and decapitated. Brains were quickly removed and placed into ice-cold, low-calcium, high-magnesium artificial cerebrospinal fluid (ACSF) with sodium replaced by choline. The medium consisted of 120 mM choline chloride, 2.5 mM KCl, 7 mM $MgSO_4$, 0.5 mM $CaCl_2$, 1.25 mM $NaH_2PO_4$, 5 mM sodium ascorbate, 3 mM sodium pyruvate, 26 mM $NaHCO_3$, and 25 mM D-(+)-glucose, and was pre-equilibrated with 95% $O_2$–5% $CO_2$. Coronal brain slices (300 μm thick) were cut with a vibratome (Leica VT1200S). Slices were incubated for 1 hr at 34°C with oxygenated ACSF containing 124 mM NaCl, 2.5 mM KCl, 2 mM $MgSO_4$, 2.5 mM $CaCl_2$, 1.25 mM $NaH_2PO_4$, 26 mM $NaHCO_3$, and 10 mM D-(+)-glucose.

## Evoked release from brain slices

Coronal brain slices (each 300 μm thick, typically with a wet weight of 17–20 mg) were transferred into a specially designed superfusion chamber with a volume of approximately 200 μl, containing freshly 95% $O_2$/5% $CO_2$ oxygenated ACSF. Slices were equilibrated for 10 min in ACSF at a superfusion rate of 0.9–1.25 ml/min. The 'control' sample was collected for 1 min just before high $K^+$ stimulation (K-ACSF, 70 mM KCl replacing equal amount of NaCl). We waited for 30 s to allow $K^+$ stimulus to immerse the slices (dead volume for solution transition of 200 μl and chamber volume of 200 μl), then the sample '70 mM K' in response to K-ACSF was collected for another 1 min. Following 10 min of washout period, we collected the third sample of 'wash' for 1 min.

To detect $Ca^{2+}$-dependent release, slices were pre-incubated for 10 min with normal ACSF and equilibrated with $Ca^{2+}$-free ACSF (containing 1 mM EGTA to chelate extracellular $Ca^{2+}$) for 10 min. The baseline sample '0 $Ca^{2+}$ ACSF' was collected for 1 min. Superfusion solution was changed to $Ca^{2+}$-free K-ACSF for 2 min and sample '0 $Ca^{2+}$ 70 mM K' was collected (dead volume for solution transition of 400 μl and chamber volume of 200 μl). Then, the solution was changed back to normal ACSF for 10 min and K-ACSF for 2 min. The sample '2.5 mM Ca 70 mM K' for the last minute was collected.

Samples were subjected to CE-MS in a method similar to SV content detection, except for the following: (1) standards were dissolved in ACSF or other buffers used in release experiment; (2) concentration ranges used for standards of Glu was from 0.003 to 1 μM; and (3) to protect the MS from salt pollution, data were collected from 10 to 20 min during CE separation.

## Patch-clamp recordings

Slices were transferred to a recording chamber on an upright fluorescent microscope equipped with differential interference contrast optics (DIC; Olympus BX51WI). Slices were submerged and superfused with ACSF at about 2.8 ml/min at 24–26°C. Whole-cell patch recordings were routinely achieved from layer 4/5 medium-sized pyramidal neurons from the somatosensory cortex. Patch pipettes (3–5 MΩ) contained 140 mM K-gluconate, 10 mM HEPES, 0.5 mM EGTA, 5 mM KCl, 3 mM $Na_2$-ATP, 0.5 mM $Na_3GTP$, and 4 mM $MgCl_2$ (with pH adjusted to 7.3 and osmolarity of 290 mOsm/kg). Current-clamp recordings were carried out with a computer-controlled amplifier (Multiclamp 700B,

Molecular Devices) and traces were digitized at 10 kHz (DigiData 1550B, Molecular Devices). Data were collected and analyzed using Clampfitor Clampex 10 software (Molecular Devices).

Cells were characterized by their membrane responses and firing patterns during hyperpolarizing and depolarizing current steps (–100 to +500 pA, increment: 50 pA or 25 pA, 500 ms). Regular spiking pyramidal neurons were identified by moderate maximal spiking frequencies (20–60 Hz, i.e., 10–30 spikes per 500 ms, *Figure 7E–G*), increasing of inter-spike intervals during depolarizing step (*Figure 7B*, *Figure 7—figure supplement 3A*), high action amplitude (*Figure 7—figure supplement 2E*), and large half width (*Figure 7—figure supplement 2F*; *Scala et al., 2019*; *Stumpf et al., 2018*). After the mean firing frequency evoked by current injections reached the steady state for at least 5 min (typically 20–30 min following the formation of whole-cell configuration), 100 µM Cr was bath-applied for 6 min. Typically, Cr was applied for a second time following washout to reconfirm the effects.

## Synaptosome preparation

Synaptosomes were isolated by Ficoll/sucrose density-gradient centrifugation (*Gulyássy et al., 2020*; *Schrimpf et al., 2005*; *Peral et al., 2010*; *Booth and Clark, 1978*). Whole brains from adult male mice were homogenized with 15 strokes at 900 rpm in buffer A (320 mM sucrose, 1 mM EDTA, 1 mM EGTA, 10 mM Tris–HCl, pH 7.4, with a complete protease inhibitor cocktail; Roche). The homogenate (H fraction) was centrifuged at 1000 × *g* for 10 min to precipitate the membrane fragments and nuclei (P1 fraction). Supernatant was centrifuged again at 1000 × *g* for 10 min, and the resulting supernatant (S1) was centrifuged at 12,000 × *g* for 20 min. Supernatant was the S2 fraction, and the pellet was resuspended with buffer A and centrifuged at 12,000 × *g* for 20 min. The resulting pellet was crude synaptosomes (P2 fraction), containing synaptosomes with mitochondria and microsomes.

Crude synaptosomes (P2 fraction) was resuspended with 150–200 µl buffer B (320 mM sucrose and 10 mM Tris–HCl [pH 7.4]). The sample was carefully overlaid on the top of a gradient of 2 ml of 7.5% (wt/vol in buffer B) Ficoll and 1.8 ml of 13% (wt/vol in buffer B) Ficoll and centrifuged at 98,000 × *g* for 45 min at 2–4°C in a swinging-bucket rotor. A myelin band was present near the surface, and the synaptosomes band (fraction Sy) was present at the interface between the 13 and 7.5% Ficoll layers, with the mitochondria being pelleted at the bottom. For further western analysis, the supernatant was divided into six fractions (600 µl for each fraction) and the mitochondria pellet was discarded. The isolated synaptosomes was included in fraction 4.

For western analysis, fractions H, S1, P1, S2, P2, and Sy were adjusted to 0.5 mg/ml by bicinchoninic acid assay (BCA) method with reference to NanoDrop 2000 Spectrophotometers. 3.35 µg protein was loaded for each lane. Fractions 1–6 were loaded with the same volume (10 µl composed of 6.7 µl sample and 3.3 µl loading buffer) for each lane.

## Creatine uptake into synaptosomes

To remove Ficoll, we diluted the synaptosomal band (480 µl) with 4.3 ml of a pH 7.4 buffer C containing (in mM) 240 mannitol, 10 glucose, 4.8 potassium gluconate, 2.2 calcium gluconate, 1.2 $MgSO_4$, 1.2 $KH_2PO_4$, and 25 HEPES-Tris. The sample was then centrifuged at 12,000 × *g*, and the pellet was resuspended with buffer C. Uptake experiments were either performed at 37°C or at 0°C (control). For each sample, 25–43 µg of synaptosomes (with a volume of 40–50 µl) were added to 360 µl buffer containing (in mmol/l) 100 NaCl, 40 mannitol, 10 glucose, 4.8 potassium gluconate, 2.2 calcium gluconate, 1.2 $MgSO_4$, 1.2 $KH_2PO_4$, 25 HEPES, and 25 Tris (pH adjusted to 7.4). A mixture of 18 µM [$^{14}$C]-creatine (0.4 µCi) and 5 µM creatine was quickly added. After 10 min, uptake was terminated by the addition of 1 ml of NaCl-free ice-cold buffer C. Samples were immediately filtered, under vacuum, through a Whatman GF/C glass filter (1825-025) pre-wetted with buffer C. Filters were further washed with 10 ml of ice-cold buffer C, dissolved in scintillation fluid, and the radioactivity determined by liquid scintillation spectrometry.

## Creatine uptake into SVs

The uptake of $^{13}$C-creatine was assayed according to a conventional procedure (*Hell et al., 1988*) with slight modifications: the immunoisolated SVs by 10 µg Syp antibody (101011, SySy) were resuspended with the uptake buffer (150 mM meglumine-tartrate, 4 mM KCl, 4 mM $MgSO_4$, 10 mM HEPES-KOH [pH 7.4], and cOmplete EDTA-free protease inhibitor cocktail) containing 4 mM Mg-ATP or additional

4 mM MgSO$_4$, followed by preincubation for 30 min at 25°C. The uptake reaction was started by addition of 1 mM $^{13}$C-creatine dissolved in the uptake buffer with a final volume of 125 µl (pH at 6.8). After 10 min at 25°C, 1 ml of ice-cold uptake buffer was added to the incubation to stop the reaction, followed by five more times washing. The SV contents were extracted using the protocol described in the determination of vesicular contents part. Then, 100 nM Cr was used as the internal control. CE-MS and LC-MS were used to verify and quantify the creatine contents of samples. A Vanquish UHPLC system coupled to a Q Exactive HF-X mass spectrometer (both instrument from Thermo Fisher Scientific) was used for LC-MS analysis along with SeQuant ZIC-HILIC column (150 mm × 2.1 mm, 3.5 µm, Merck Millipore, 150442) in the positive mode and SeQuant ZIC-pHILIC column (150 mm × 2.1 mm, 5 µm, Merck Millipore, 150460) in the negative mode. For ZIC-HILIC column, the mobile phase A was 0.1% formic acid in water and the mobile phase B was 0.1% formic acid in acetonitrile. The linear gradient was as follows: 0 min, 80% B; 6 min, 50% B; 13 min, 50% B; 14 min, 20% B; 18 min, 20% B; 18.5 min, 80% B; and 30 min, 80% B. The flow rate used was 300 µl/min and the column temperature was maintained at 30°C. For ZIC-pHILIC column, the mobile phase A is 20 mM ammonium carbonate in water, adjusted to pH 9.0 with 0.1% ammonium hydroxide solution (25%), and the mobile phase B is 100% acetonitrile. The linear gradient was as follows: 0 min, 80% B; 2 min, 80% B; 19 min, 20% B; 20 min, 80% B; and 30 min, 80% B. The flow rate used was 150 µl/min, and the column temperature was 25°C. Samples were maintained at 4°C in Vanquish autosampler. Then, 3 µl of extracted metabolites were injected for each run. IP samples were subjected to ZIC-HILIC column in positive mode for major metabolites detection, and then subject to ZIC-pHILIC column in negative mode for orthogonal detection.

## Acknowledgements

We are grateful to Drs. Xiaohui Zhang, Xinxiang Zhang, Minmin Luo, Qingchun Guo, Yiqun Liu, Yingchun Hu, and Wuping Ge for discussion, suggestions, and technical support; Jiang Chen, Jing Cai, Yulin Jiang, Jinhuan Ou, Xiaomeng Deng, Jiawen Liu, and Meng Wu for technical assistance; and CIBR, Peking-Tsinghua Center for Life Sciences, Changping Laboratory, Chinese Academy of Medical Sciences (2019RU003), and National Natural Science Foundation of China (Project 32061143017 to YR and 81473189) for support. Research in the Rao lab has never been contaminated by the Chinese Brain Initiative.

## Additional information

### Funding

| Funder | Grant reference number | Author |
|---|---|---|
| Chinese Academy of Medical Sciences | 2019RU003 | Yi Rao |
| Chinese Institute for Brain Research | | Yi Rao |
| Peking-Tsinghua Center for Life Sciences | | Yi Rao |
| Changping Laboratory | | Yi Rao |
| National Natural Science Foundation of China | 32061143017 | Yi Rao |
| National Natural Science Foundation of China | 81473189 | Xiling Bian |

The funders had no role in study design, data collection and interpretation, or the decision to submit the work for publication.

### Author contributions

Xiling Bian, Data curation, Formal analysis, Funding acquisition, Validation, Investigation, Visualization, Methodology, Writing – original draft, Writing – review and editing, designed the experiments,

improved the methods, carried out experiments and supervised technicians to carry out the experiments for this paper; Jiemin Zhu, Data curation, Formal analysis, Validation, Investigation, Methodology, Writing – original draft; Xiaobo Jia, Data curation, Formal analysis, Validation, Investigation, Visualization, Methodology, Writing – original draft, Writing – review and editing, designed the experiment and reproducibly found Cr in SVs; Wenjun Liang, Data curation, Formal analysis, Validation, Investigation, Visualization, Methodology; Sihan Yu, Data curation, Formal analysis, Validation, Investigation, Visualization; Zhiqiang Li, Data curation, Investigation; Wenxia Zhang, Writing – review and editing; Yi Rao, Conceptualization, Resources, Formal analysis, Supervision, Funding acquisition, Investigation, Writing – original draft, Project administration, Writing – review and editing, YR conceived the idea of searching for new neurotransmitters and supervised the project

## Author ORCIDs
Xiaobo Jia http://orcid.org/0000-0002-4214-8906
Yi Rao https://orcid.org/0000-0002-0405-5426

## Ethics
All animal procedures were approved by the Animal Center of Peking University and Animal Care. Experiments were carried out in accordance with the guidelines of Institutional Animal Care and Use Committee (IACUC) of Peking University (LSC-RaoY-9 and LSC-RaoY-10).

Reviewer #1 (Public Review): https://doi.org/10.7554/eLife.89317.4.sa1
Reviewer #2 (Public Review): https://doi.org/10.7554/eLife.89317.4.sa2
Reviewer #3 (Public Review): https://doi.org/10.7554/eLife.89317.4.sa3
Author Response https://doi.org/10.7554/eLife.89317.4.sa4

---

# Additional files

## Supplementary files
• MDAR checklist

## Data availability
All data generated or analyzed during this study are included in the manuscript and supporting file.

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
