## [Editor Report · eLife assessment]

This study presents **valuable** observations on a potential role of creatine (Cr) as a novel neurotransmitter. The data provide **solid** evidence that Cr is present in synaptic vesicles. If, in the future, a receptor can be described, it will support the claim that Cr is synaptically released and binds to a post-synaptic receptor. This would be of wide interest to the field of neuroscience.

---

## [Referee Report · Reviewer #1 (Public Review)]

The overall tone of the rebuttal and lack of responses on several questions was surprising. Clearly, the authors did not appreciate the phrase 'no smoking gun' and provided a lengthy repetition of the fair argument about 'ticking boxes' on the classic list of criteria. They also make repeated historical references that descriptions of neurotransmitters include many papers, typically over decades, e.g. in the case of ACh and its discovery by Sir Henry Dale. While I empathize with the authors' apparent frustration (I quote: '...accept the reality that Rome was not built in a single day and that no transmitter was proven by a one single paper') I am a bit surprised at the complete brushing away of the argument, and in fact the discussion. In the original paper, the notion of a receptor was mentioned only in a single sentence and all three reviewers brought up this rather obvious question. The historical comparisons are difficult: Of course many papers contribute to the identification of a neurotransmitter, but there is a much higher burden of proof in 2023 compared to the work by Otto Loewi and Sir Henry Dale: most, if not all, currently accepted neurotransmitter have a clear biological function at the level of the brain and animal behavior or function - and were in fact first proposed to exist based on a functional biological experiment (e.g. Loewi's heart rate change). This, and the isolation of the chemical that does the job, were clear, unquestionable 'smoking guns' a hundred years ago. Fast forward 2023: Creatine has been carefully studied by the authors to tick many of the boxes for neurotransmitters, but there is no clear role for its function in an animal. The authors show convincing effects upon K+ stimulation and electrophysiological recordings that show altered neuronal activity using the slc6a8 and agat mutants as well as Cr application - but, as has been pointed out by other reviewers, these effects are not a clear-cut demonstration of a chemical transmitter function, however many boxes are ticked. The identification of a role of a neurotransmitter for brain function and animal behavior has reasonably more advanced possibilities in 2023 than a hundred years ago - and e.g. a discussion of approaches for possible receptor candidates should be possible.

Again, I reviewed this positively and agree that a lot of cumulative data are great to be put out there and allow the discovery to be more broadly discussed and tested. But I have to note, that the authors simply respond with the 'Rome was not built in a single day' statement to my suggestions on at least 'have some lead' how to approach the question of a receptor e.g. through agonists or antagonists (while clearly stating 'I do not think the publication of this manuscript should not be made dependent' on this). Similarly, in response to reviewer 2's concerns about a missing receptor, the authors' only (may I say snarky) response is ' We have deleted this sentence, though what could mediate postsynaptic responses other than receptors?' The bullet point by reviewer 3 ' • No candidate receptor for creatine has been identified postsynaptically.' is the one point by that reviewer that is simply ignored by the authors completely. Finally, I note that my reivew question on the K stimulation issues (e.g. 35 neurons that simply did not respond at all) was: ' Response: To avoid the disadvantage of K stimulation, we also performed optogenetic experiments recently and obtained encouraging preliminary results.' No details, not data - no response really.

In sum, I find this all a bit strange and the rebuttal surprising - all three reviewers were supportive and have carefully listed points of discussion that I found all valid and thoughtful. In response, the authors selectively responded scientifically to some experimental questions, but otherwise simply rather non-scientifically dismissed questions with 'Rome was not built in a day'-type answers, or less. I my view, the authors have disregarded the review process and the effort of three supportive reviewers, which should be part of the permanent record of this paper.

---

## [Referee Report · Reviewer #2 (Public Review)]

Summary:

Bian et al studied creatine (Cr) in the context of central nervous system (CNS) function. They detected Cr in synaptic vesicles purified from mouse brains with anti-Synaptophysin using capillary electrophoresis-mass spectrometry. Cr levels in the synaptic vesicle fraction was reduced in mice lacking the Cr synthetase AGAT, or the Cr transporter SLC6A8. They provide evidence for Cr release within several minutes after treating brain slices with KCl. This KCl-induced Cr release was partially calcium dependent and was attenuated in slices obtained from AGAT and SLC6A8 mutant mice. Cr application also decreased the excitability of cortical pyramidal cells in one third of the cells tested. Finally, they provide evidence for SLC6A8-dependent Cr uptake into synaptosomes, and ATP-dependent Cr loading into synaptic vesicles. Based on these data, the authors propose that Cr may act as neurotransmitter in the CNS.

Strengths:

A major strength of the paper is the broad spectrum of tools used to investigate Cr.The study provides evidence that Cr is present in/loaded into synaptic vesicles.

Weaknesses (resubmission):

There is no significant decrease in Cr content pulled down by anti-Syp in AGAT-/- mice when normalized to IgG controls. Hence, blocking AGAT activity/Cr synthesis does not affect Cr levels in the synaptic vesicle fraction, arguing against a Cr enrichment.There is no difference in KCl-induced Cr release between SLC6A8-/Y and SLC6A8+/Y when normalizing the data to the respective controls. Thus, the data are not consistent with the idea that depolarization-induced Cr release requires SLC6A8.The rationale of grouping the excitability data into responders and non-responders is not convincing because the threshold of 10% decrease in AP rate is arbitrary. The data do therefore not support the conclusion that Cr reduces neuronal excitability.

---

## [Referee Report · Reviewer #3 (Public Review)]

SUMMARY:

The manuscript by Bian et al. promotes the idea that creatine is a new neurotransmitter. The authors conduct an impressive combination of mass spectrometry (Fig. 1), genetics (Figs. 2, 3, 6), biochemistry (Figs. 2, 3, 8), immunostaining (Fig. 4), electrophysiology (Figs. 5, 6, 7), and EM (Fig. 8) in order to offer support for the hypothesis that creatine is a CNS neurotransmitter.

STRENGTHS:

There are many strengths to this study.

The combinatorial approach is a strength. There is no shortage of data in this study.The careful consideration of specific criteria that creatine would need to meet in order to be considered a neurotransmitter is a strength.The comparison studies that the authors have done in parallel with classical neurotransmitters are helpful.Demonstration that creatine has inhibitory effects is another strength.The new genetic mutations for Slc6a8 and AGAT are strengths and potentially incredibly helpful for downstream work.

WEAKNESSES:

Some data are indirect. Even though Slc6a8 and AGAT are helpful sentinels for the presence of creatine, they are not creatine themselves. Of note, these molecules themselves are not essential for making the case that creatine is a neurotransmitter.Regarding Slc6a8, it seems to work only as a reuptake transporter - not as a transporter into SVs. Therefore, we do not know what the transporter into the TVs is.Puzzlingly, Slc6a8 and AGAT are in different cells, setting up the complicated model that creatine is created in one cell type and then processed as a neurotransmitter in another. This matter will likely need to be resolved in future studies.No candidate receptor for creatine has been identified postsynaptically. This will likely need to be resolved in future studies.Because no candidate receptor has been identified, it is important to fully consider other possibilities for roles of creatine that would explain these observations other than it being a neurotransmitter? There is some attention to this in the Discussion.

There are several criteria that define a neurotransmitter. The authors nicely delineated many criteria in their discussion, but it is worth it for readers to do the same with their own understanding of the data.

By this reviewer's understanding (and combining some textbook definitions together) a neurotransmitter: 1) must be present within the presynaptic neuron and stored in vesicles; 2) must be released by depolarization of the presynaptic terminal; 3) must require Ca2+ influx upon depolarization prior to release; 4) must bind specific receptors present on the postsynaptic cell; 5) exogenous transmitter can mimic presynaptic release; 6) there exists a mechanism of removal of the neurotransmitter from the synaptic cleft.

For a paper to claim that the published work has identified a new neurotransmitter, several of these criteria would be met - and the paper would acknowledge in the discussion which ones have not been met. For this particular paper, this reviewer finds that condition 1 is clearly met.

Conditions 2 and 3 seem to be met by electrophysiology, but there are caveats here. High KCl stimulation is a blunt instrument that will depolarize absolutely everything in the prep all at once and could result in any number of non-specific biological reactions as a result of K+ rushing into all neurons in the prep. Moreover, the results in 0 Ca2+ are puzzling. For creatine (and for the other neurotransmitters), why is there such a massive uptick in release, even when the extracellular saline is devoid of calcium?

Condition 4 is not discussed in detail at all. In the discussion, the authors elide the criterion of receptors specified by Purves by inferring that the existence of postsynaptic responses implies the existence of receptors. True, but does it specifically imply the existence of creatinergic receptors? This reviewer does not think that is necessarily the case. The authors should be appropriately circumspect and consider other modes of inhibition that are induced by activation or potentiation of other receptors (e.g., GABAergic or glycinergic).

Condition 5 may be met, because authors applied exogenous creatine and observed inhibition. However, this is tough to know without understanding the effects of endogenous release of creatine. if they were to test if the absence of creatine caused excess excitation (at putative creatinergic synapses), then that would be supportive of the same. Nicely, Ghirardini et al., 2023 study cited by the reviewers does provide support for this exact notion in pyramidal neurons.

For condition 6, the authors made a great effort with Slc6a8. This is a very tough criterion to understand or prove for many synapses and neurotransmitters.

In terms of fundamental neuroscience, the story should be impactful. There are certainly more neurotransmitters out there than currently identified and by textbook criteria, creatine seems to be one of them taking all of the data in this study and others into account.

---

## [Author Response]

The following is the authors’ response to the previous reviews:

Point-to-Point Responses to Reviewers’ Comments

We are a bit surprised by the comments of Reviewer 1, but that our further responses can help communications with Reviewer 1. We have also responded to comments of Reviewers 2 and 3.

**Reviewer #1 (Public Review):**
The overall tone of the rebuttal and lack of responses on several questions was surprising. Clearly, the authors took umbrage at the phrase 'no smoking gun' and provided a lengthy repetition of the fair argument about 'ticking boxes' on the classic list of criteria. They also make repeated historical references that descriptions of neurotransmitters include many papers, typically over decades, e.g. in the case of ACh and its discovery by Sir Henry Dale. While I empathize with the authors' apparent frustration (I quote: '...accept the reality that Rome was not built in a single day and that no transmitter was proven by a one single paper') I am a bit surprised at the complete brushing away of the argument, and in fact the discussion. In the original paper, the notion of a receptor was mentioned only in a single sentence and all three reviewers brought up this rather obvious question. The historical comparisons are difficult: Of course many papers contribute to the identification of a neurotransmitter, but there is a much higher burden of proof in 2023 compared to the work by Otto Loewi and Sir Henry Dale: most, if not all, currently accepted neurotransmitter have a clear biological function at the level of the brain and animal behavior or function - and were in fact first proposed to exist based on a functional biological experiment (e.g. Loewi's heart rate change). This, and the isolation of the chemical that does the job, were clear, unquestionable 'smoking guns' a hundred years ago. Fast forward 2023: Creatine has been carefully studied by the authors to tick many of the boxes for neurotransmitters, but there is no clear role for its function in an animal. The authors show convincing effects upon K+ stimulation and electrophysiological recordings that show altered neuronal activity using the slc6a8 and agat mutants as well as Cr application - but, as has been pointed out by other reviewers, these effects are not a clear-cut demonstration of a chemical transmitter function, however many boxes are ticked. The identification of a role of a neurotransmitter for brain function and animal behavior has reasonably more advanced possibilities in 2023 than a hundred years ago - and e.g. a discussion of approaches for possible receptor candidates should be possible.Again, I reviewed this positively and agree that a lot of cumulative data are great to be put out there and allow the discovery to be more broadly discussed and tested. But I have to note, that the authors simply respond with the 'Rome was not built in a single day' statement to my suggestions on at least 'have some lead' how to approach the question of a receptor e.g. through agonists or antagonists (while clearly stating 'I do not think the publication of this manuscript should not be made dependent' on this). Similarly, in response to reviewer 2's concerns about a missing receptor, the authors' only (may I say snarky) response is ' We have deleted this sentence, though what could mediate postsynaptic responses other than receptors?' The bullet point by reviewer 3 ' • No candidate receptor for creatine has been identified postsynaptically.' is the one point by that reviewer that is simply ignored by the authors completely. Finally, I note that my reivew question on the K stimulation issues (e.g. 35 neurons that simply did not respond at all) was: ' Response: To avoid the disadvantage of K stimulation, we also performed optogenetic experiments recently and obtained encouraging preliminary results.' No details, not data - no response really.In sum, I find this all a bit strange and the rebuttal surprising - all three reviewers were supportive and have carefully listed points of discussion that I found all valid and thoughtful. In response, the authors selectively responded scientifically to some experimental questions, but otherwise simply rather non-scientifically dismissed questions with 'Rome was not built in a day'-type answers, or less. I my view, the authors have disregarded the review process and the effort of three supportive reviewers, which should be part of the permanent record of this paper.

Response:

We were very surprised by the tone of Reviewer 1 in the second round of reviewing. The corresponding author has spent some time including a long holiday to cool down and re-read our earlier responses. The following is entirely by the corresponding author.

I have finally checked the term “smoking gun”, and found out that I interpreted it wrongly while I had thought that Reviewer 1 was wrong. This came from a long story in that I was lectured by a native speaker for my English when submitting the first paper from my own paper. In that case, the Reviewer was wrong (in arguing that only adjectives but not nouns can be used to define nouns), I was quite offended and remembered it vividly. In the case of “smoking gun”, I wrongly believed that it meant a hint (while the definite evidence would be “the final nail in the coffin”). By interpreting is as a hint, I was then rebutting Reviewer 1 for negating all our experimental results as “not a single piece of suggestive evidence”.

For the above, I apologize.

I have another disagreement about “smoking gun”. For a transmitter, multiple criteria have to be met. For example, finding a receptor for a small molecule would not be definitive for a transmitter because if it is not present in the SVs, it is unlikely to be a typical transmitter. If a molecule has a receptor but they are not even in the nervous system, it is definitely no a transmitter.

The title of our paper is “Evidence suggesting creatine as a new central neurotransmitter”, not “Evidence proving creatine as a new central neurotransmitter”. In the Abstract, after “Our biochemical, chemical, genetic and electrophysiological results are consistent with the possibility of Cr as a neurotransmitter”, we are adding “though not yet reaching the level of proof for the now classic transmitters”. In the last sentence of the introduction, we have now added “though the discovery of a receptor for Cr would prove it”.

I do, however, believe that, however strong the wordings are, criticisms and rebuttals in science are normal and should be conducted even when emotions are involved.

One of my major point of differences with at least two of the reviewers is that the criteria for neurotransmitters should be those listed in major textbooks. While everyone can have one’s own opinions, the textbooks, especially those accepted by readers of the field for more than 40 years, should be the standards. Kandel has listed the 4 criteria not only 40 years ago but also just 2 years ago in their latest 6th edition. The reviewers have asked for more, while discounting Kandel et al. (2021). So, in essence, the Reviewer is not shy in scientific criticisms when stating “The identification of a role of a neurotransmitter for brain function and animal behavior has reasonably more advanced possibilities in 2023 than a hundred years ago”.

Reviewer 1 raised another new criterion: brain function and behavior, while this is not in any textbook lists. However, lack of Cr caused behavioral problems, as cited by us in the introduction: both humans and mice were defective in brain function with loss of function mutations in the gene for the specific Cr transporter SLC6A8. If the reviewer meant behavioral abnormalities caused by Cr injection, that was unclear. But that criterion may not be met by other transmitters which is the likely reason that it was not a criterion in any textbook.

**Reviewer #2 (Public Review):**
Summary:Bian et al studied creatine (Cr) in the context of central nervous system (CNS) function. They detected Cr in synaptic vesicles purified from mouse brains with anti-Synaptophysin using capillary electrophoresis-mass spectrometry. Cr levels in the synaptic vesicle fraction was reduced in mice lacking the Cr synthetase AGAT, or the Cr transporter SLC6A8. They provide evidence for Cr release within several minutes after treating brain slices with KCl. This KCl-induced Cr release was partially calcium dependent and was attenuated in slices obtained from AGAT and SLC6A8 mutant mice. Cr application also decreased the excitability of cortical pyramidal cells in one third of the cells tested. Finally, they provide evidence for SLC6A8-dependent Cr uptake into synaptosomes, and ATP-dependent Cr loading into synaptic vesicles. Based on these data, the authors propose that Cr may act as neurotransmitter in the CNS.Strengths:1. A major strength of the paper is the broad spectrum of tools used to investigate Cr.2. The study provides evidence that Cr is present in/loaded into synaptic vesicles.Weaknesses:1. There is no significant decrease in Cr content pulled down by anti-Syp in AGAT-/- mice when normalized to IgG controls. Hence, blocking AGAT activity/Cr synthesis does not affect Cr levels in the synaptic vesicle fraction, arguing against a Cr enrichment.

Response: Evidence for Cr enrichment in the SVS was obtained robustly with wild type mice. When brain Cr is very low in AGAT-/- mutant mice, because there is little Cr, there is also little Cr in the SVs. One does not require that as a criterion: it does not argue against the normal levels of Cr could be transported into the SVs even if when the much reduced levels of AGAT-/- Cr in mutant mice could be enriched in SVs.

2. There is no difference in KCl-induced Cr release between SLC6A8-/Y and SLC6A8+/Y when normalizing the data to the respective controls. Thus, the data are not consistent with the idea that depolarization-induced Cr release requires SLC6A8.

Response: This comment of Reviewer 2 was based on Figure 5D. But if one carefully examines Figure 5G, it was clear that the Ca++ dependent component of KCl -induced Cr release was lower in SLC6A8-/Y than that in SLC6A8+/Y.

3. The rationale of grouping the excitability data into responders and non-responders is not convincing because the threshold of 10% decrease in AP rate is arbitrary. The data do therefore not support the conclusion that Cr reduces neuronal excitability.

Response: Comparison of the same neuron, before and after Cr did show effects on neuronal excitability though that would have no statistics if one does not group multiple cells into the same categories.

**Reviewer #3 (Public Review):**
SUMMARY:The manuscript by Bian et al. promotes the idea that creatine is a new neurotransmitter. The authors conduct an impressive combination of mass spectrometry (Fig. 1), genetics (Figs. 2, 3, 6), biochemistry (Figs. 2, 3, 8), immunostaining (Fig. 4), electrophysiology (Figs. 5, 6, 7), and EM (Fig. 8) in order to offer support for the hypothesis that creatine is a CNS neurotransmitter.STRENGTHS:There are many strengths to this study.• The combinatorial approach is a strength. There is no shortage of data in this study.• The careful consideration of specific criteria that creatine would need to meet in order to be considered a neurotransmitter is a strength.• The comparison studies that the authors have done in parallel with classical neurotransmitters are helpful.• Demonstration that creatine has inhibitory effects is another strength.• The new genetic mutations for Slc6a8 and AGAT are strengths and potentially incredibly helpful for downstream work.WEAKNESSES:• Some data are indirect. Even though Slc6a8 and AGAT are helpful sentinels for the presence of creatine, they are not creatine themselves. Of note, these molecules themselves are not essential for making the case that creatine is a neurotransmitter.

Response: We agree, but those data are not inconsistent with the possibility.

• Regarding Slc6a8, it seems to work only as a reuptake transporter - not as a transporter into SVs. Therefore, we do not know what the transporter into the TVs is.

Response: SLC6A8 is not the transporter on the SVs, but is an excellent candidate for the transporter on the presynaptic cytoplasmic membrane for uptake of Cr into the presynaptic structure.

• Puzzlingly, Slc6a8 and AGAT are in different cells, setting up the complicated model that creatine is created in one cell type and then processed as a neurotransmitter in another. This matter will likely need to be resolved in future studies.

Response: We agree.

• No candidate receptor for creatine has been identified postsynaptically. This will likely need to be resolved in future studies.

Response: We agree.

• Because no candidate receptor has been identified, it is important to fully consider other possibilities for roles of creatine that would explain these observations other than it being a neurotransmitter? There is some attention to this in the Discussion.

Response: We agree.

There are several criteria that define a neurotransmitter. The authors nicely delineated many criteria in their discussion, but it is worth it for readers to do the same with their own understanding of the data.By this reviewer's understanding (and combining some textbook definitions together) a neurotransmitter: 1) must be present within the presynaptic neuron and stored in vesicles; 2) must be released by depolarization of the presynaptic terminal; 3) must require Ca2+ influx upon depolarization prior to release; 4) must bind specific receptors present on the postsynaptic cell; 5) exogenous transmitter can mimic presynaptic release; 6) there exists a mechanism of removal of the neurotransmitter from the synaptic cleft.

Response: While any of us can come up with a list according to our own understanding, the paper copies lists from textbooks, especially from Kandel et al. (2021), which lists the same 4 criteria as Kandel et al. (1983), providing consistency and consensus.

For a paper to claim that the published work has identified a new neurotransmitter, several of these criteria would be met - and the paper would acknowledge in the discussion which ones have not been met. For this particular paper, this reviewer finds that condition 1 is clearly met.Conditions 2 and 3 seem to be met by electrophysiology, but there are caveats here. High KCl stimulation is a blunt instrument that will depolarize absolutely everything in the prep all at once and could result in any number of non-specific biological reactions as a result of K+ rushing into all neurons in the prep. Moreover, the results in 0 Ca2+ are puzzling. For creatine (and for the other neurotransmitters), why is there such a massive uptick in release, even when the extracellular saline is devoid of calcium?

Response: Classic transmitters are released in a Ca++ dependent manner when stimulated by KCl, though they also had a Ca++ independent component as also shown in our Figure 5 E and F.

Condition 4 is not discussed in detail at all. In the discussion, the authors elide the criterion of receptors specified by Purves by inferring that the existence of postsynaptic responses implies the existence of receptors. True, but does it specifically imply the existence of creatinergic receptors? This reviewer does not think that is necessarily the case. The authors should be appropriately circumspect and consider other modes of inhibition that are induced by activation or potentiation of other receptors (e.g., GABAergic or glycinergic).

Response: Kandel et al. did not list this.

Condition 5 may be met, because authors applied exogenous creatine and observed inhibition. However, this is tough to know without understanding the effects of endogenous release of creatine. if they were to test if the absence of creatine caused excess excitation (at putative creatinergic synapses), then that would be supportive of the same. Nicely, Ghirardini et al., 2023 study cited by the reviewers does provide support for this exact notion in pyramidal neurons.

Response: For most commonly accepted transmitters, this criterion has never been met. For example, the simplest case would be ACh at the neuromuscular junction. Howver, we have now found that choline is clearly present in SVs. So, how does anyone be sure that only ACh is released only, or how does anyone rule out effects of choline on postsynaptic cells when cholinergic neurons are stimulated?

Many synapses are now known to release more than one transmitter, making it difficult to define the effect of one transmitter released endogenously.

These are perhaps reasons why some textbooks do not emphasize similarities of endogenously released vs exogenously applied molecules.

For condition 6, the authors made a great effort with Slc6a8. This is a very tough criterion to understand or prove for many synapses and neurotransmitters.

Response: SLC6A8 is a transporter on the cytoplasmic membrane, thus a good candidate for removal of Cr from the synaptic cleft.

In terms of fundamental neuroscience, the story should be impactful. There are certainly more neurotransmitters out there than currently identified and by textbook criteria, creatine seems to be one of them taking all of the data in this study and others into account.

Response: We hope that more will join our lonely efforts in trying to discover more transmitters.

**Reviewer #1 (Recommendations For The Authors):**
Since the authors largely disregarded questions in the review process, I do not see a point in listing recommendation for the authors again.

**Reviewer #2 (Recommendations For The Authors):**
1. The different sections of the manuscript are not separated by headers.

Response: We do have separate subheadings.

2. The beginning of the results section either does not reference the underlying literature or refers to unpublished data.

Response: We have a very long introduction which was criticized for being too long and with too much historical citations. We therefore refrained from citation again in the beginning part of the Results section.

3. The text contains many opinions and historical information that are not required (e.g., "It has never been easy to discover a new neurotransmitter, especially one in the central nervous system (CNS). We have been searching for new neurotransmitters for 12 years."; l. 17).

Response: We would like to keep these because most readers are young and do not know the history and difficulties of discovering transmitters.

4. Almeida et al. (2008; doi: 10.1002/syn.20280) provided evidence for electrical activity-, and Ca2+-dependent Cr release from rat brain slices. This paper should be introduced in the introduction.

Response: Done.

5. Fig. 7: A Y-scale for the stimulation protocol is missing.

Response: Done.

**Reviewer #3 (Recommendations For The Authors):**
The main suggestion by this reviewer (beyond the details in the public review) was to consider the full spectrum of biology that is consistent with these results. By my reading, creatine could be a neurotransmitter, but other possibilities also exist. The authors have highlighted some of those for their Discussion.